# A IS FOR ABSORPTION: STUDYING FEATURE SPLITTING AND ABSORPTION IN SPARSE AUTOENCODERS

## ABSTRACT

Sparse Autoencoders (SAEs) have emerged as a promising approach to decompose the activations of Large Language Models (LLMs) into human-interpretable latents. In this paper, we pose two questions. First, to what extent do SAEs extract monosemantic and interpretable latents? Second, to what extent does varying the sparsity or the size of the SAE affect monosemanticity / interpretability? By investigating these questions in the context of a simple first-letter identification task where we have complete access to ground truth labels for all tokens in the vocabulary, we are able to provide more detail than prior investigations. Critically, we identify a problematic form of feature-splitting we call "feature absorption" where seemingly monosemantic latents fail to fire in cases where they clearly should. Our investigation suggests that varying SAE size or sparsity is insufficient to solve this issue, and that there are deeper conceptual issues in need of resolution. We release a feature absorption explorer at `https://feature-absorption.streamlit.app`.

## 1 INTRODUCTION

Large Language Models (LLMs) have achieved remarkable performance across a wide range of tasks, yet our understanding of their internal mechanisms lags behind their capabilities. This gap between performance and interpretability raises concerns about the "black box" nature of these models (Rudin, 2019). The field of mechanistic interpretability aims to address this issue by reverse-engineering the internal algorithms of neural networks and performing causal analysis on them (Olah et al., 2020).

One recent promising approach in this field is the use of Sparse Autoencoders (SAEs), which have shown potential in decomposing the dense, polysemantic activations of LLMs into more "interpretable" latent features (Cunningham et al., 2024; Bricken et al., 2023) using sparse dictionary learning (Olshausen & Field, 1997). SAE neurons (hereafter called "latents")[1] are said to be interpretable if they appear to detect some property of the input (which we refer to as a "feature") and classify that feature with high precision / recall (Bricken et al., 2023).

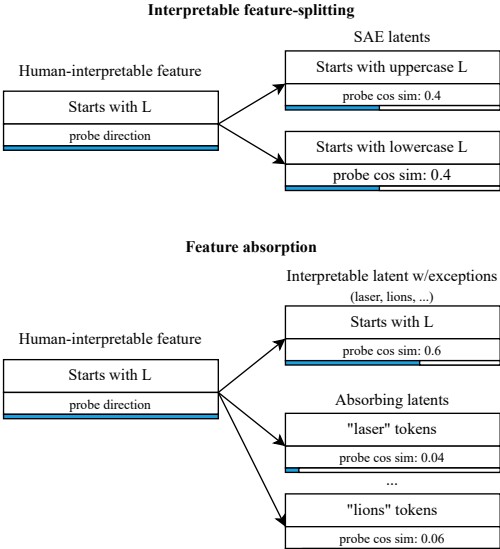

Figure 1: In feature absorption, an SAE latent appears interpretable, but has arbitrary exceptions where different latents "absorb" the feature direction and activate instead. The absorbing latents are frequently, though not always, token-aligned.

---

[1] We use *latents* to prevent overloading the term *feature*, which we reserve for human-interpretable concepts the SAE may capture. This breaks from earlier usage which used *feature* for both (Elhage et al., 2022), but aligns with the terminology in (Lieberum et al., 2024) and makes the distinction more clear.

However, despite these theoretical advantages, most existing work on SAE interpretability mainly studies max activating examples (Cunningham et al., 2024), which may be misleading. There are more rigorous works which only measure precision (Bricken et al., 2023; Templeton et al., 2024; Kissane et al., 2024). Recent work has briefly explored recall and found it to be worse than naively expected, but this remains poorly understood (Olah et al., 2024b). We build on this work by evaluating precision / recall on a large number of SAEs, and offer a partial explanation for lower-than-expected recall of SAE latents in the form of "feature absorption".

Our key contributions in this investigation include the following:

1. We identify numerous SAE latents appearing to classify first-letter features. We calculate their precision / recall on the first letter identification tasks (as a proxy for monosemanticity / interpretability) and find they significantly underperform linear probes.

2. We find that latents which seem outwardly to classify the same feature can have vastly different precision / recall and that this tradeoff is mediated by various factors, mainly sparsity and width of the SAE.

3. Most importantly, we identify and quantify a variant of feature-splitting we call "feature absorption", where an SAE latent appears to track a human-interpretable concept, but it fails to activate on seemingly arbitrary tokens. Instead, a different latent activates and contributes a portion of the probe direction, "absorbing" the feature. This is described in Figure 1.

We believe that feature absorption poses an obstacle to the practical application of SAEs since it suggests SAE latents may be inherently unreliable classifiers. This is particularly important if we seek to use them in safety applications where we need confidence that latents are fully tracking behaviours, such as bias or deceptive behavior. Furthermore, techniques which seek to describe circuits in terms of a sparse combination of latents with also be more difficult in the context of feature absorption (Marks et al., 2024).

## 2 BACKGROUND

**Linear probing.** A linear probe is a simple linear classifier trained on the hidden activations of a neural network, typically using logistic regression (LR) (Alain & Bengio, 2017).

**K-sparse probing.** A k-sparse probe (Gurnee et al., 2023) is a linear probe trained on a sparse subset of $k$ neurons or SAE latents. Training a k-sparse probe first requires selecting the $k$ best neurons or SAE latents that in-aggregate act as a good classifier, and then training a standard linear probe on just those $k$ neurons or latents.

Gurnee et al. (2023) proposed several methods of estimating the best $k$ neurons or features to pick, one of which involves first training a LR probe with a L1 loss term, and selecting the $k$ largest elements by probe weight. When we refer to k-sparse probing in this work, we use this method of selecting $k$ features.

**Sparse autoencoders.** An SAE consists of an encoder, $W_{enc}$, a decoder, $W_{dec}$, and corresponding biases $b_{enc}$ and $b_{dec}$. The SAE has a nonlinearity, $\sigma$, typically a ReLU (or variants such as JumpReLU (Rajamanoharan et al., 2024; Lieberum et al., 2024)). Given input activation, $a$, the SAE computes a hidden representation, $f$, and reconstruction, $\hat{a}$:

$$f = \sigma(W_{enc}a + b_{enc}) \tag{1}$$
$$\hat{a} = W_{dec}f + b_{dec} \tag{2}$$

SAEs attempt to reconstruct input activations by projecting into an overcomplete basis using a sparsity-inducing loss term (typically $L1$ loss), or a certain number of non-zero features ($L0$) on the hidden activations. SAEs learn feature decompositions in an unsupervised manner, and while the sparsity penalty is meant to encourage monosemantic features, it is often hard to judge if the features learned are interpretable or to say with certainty that the features the SAE learned are faithful to the computation performed by the underlying LLM.

**SAE feature ablation.** We often want to understand how an SAE latent causally influences a downstream output. In an ablation study, the latent in question is removed from the computation graph of the model to see the effect this has on a downstream metric. A negative ablation effect means removing the SAE latent would lower the metric.

We follow the work of Marks et al. (2024) and provide the procedure in Algorithm 1 below. We also make use of the integrated-gradients (IG) approximation (Sundararajan et al., 2017) to improve the speed of running multiple ablation experiments.

---

**Algorithm 1** SAE Latent Ablation

---

1: Insert SAE in model computation path, including error term
2: Define a scalar metric on the model's output distribution (e.g. difference between token logits)
3: Calculate baseline metric value for a test prompt
4: **for** each token of interest **do**
5:    **for** each SAE latent **do**
6:       Set the SAE latent activation to 0
7:       Recalculate the metric
8:       Compute ablation effect as (baseline metric - new metric)
9:       Reset the SAE latents to its original value
10:    **end for**
11: **end for**

---

## 3 EXPERIMENTAL SETUP

Our experiments focused on predicting the first-letter of a single token containing characters from the English alphabet (a-z, A-Z) and an optional leading space. We use in-context learning (ICL) prompts to elicit knowledge from the model, using templates of the form:

```
{token} has the first letter: {capitalized_first_letter}
```

An example of an ICL prompt consisting of 2 in-context examples is shown below. The model should output the _D token:

```
tartan has the first letter: T
mirth has the first letter: M
dog has the first letter:
```

In the above prompt, we extract residual stream activations at the _dog token index. These activations are used both for LR probe training and for applying SAEs. We use a train/test split of 80% / 20%, and evaluate only on the test set, including when running experiments on SAEs. When applying SAEs, we include the SAE error term (Marks et al., 2024) to avoid changing model output.

To determine the causal effect of SAE latents on the first-letter identification task we conduct ablation studies. We use a metric consisting of the logit of the correct letter minus the mean logit of all incorrect letters. This measures the propensity of the model to choose the correct starting letter as opposed to other letters. Formally, our metric $m$ is defined below, where $g$ refers to the final token logits, $L$ is the set of uppercase letters, and $y$ is the uppercase letter that is the correct starting letter:

$$m = g[y] - \frac{1}{|L| - 1} \sum_{l \in \{L \setminus y\}} g[l]$$

We discuss this metric and alternative formulations further in Appendix A.8.

To determine how well multiple features perform as a classifier when used together, we use k-sparse probing, increasing the value of $k$ from 1 to 15. We train a LR probe using a L1 loss term with coefficient 0.01, and select the top $k$ features by magnitude.

We use the base Gemma-2-2B model for most of our studies, along with the full set of Gemma Scope residual stream SAEs of width 16k and 65k released by Deepmind (Lieberum et al., 2024).

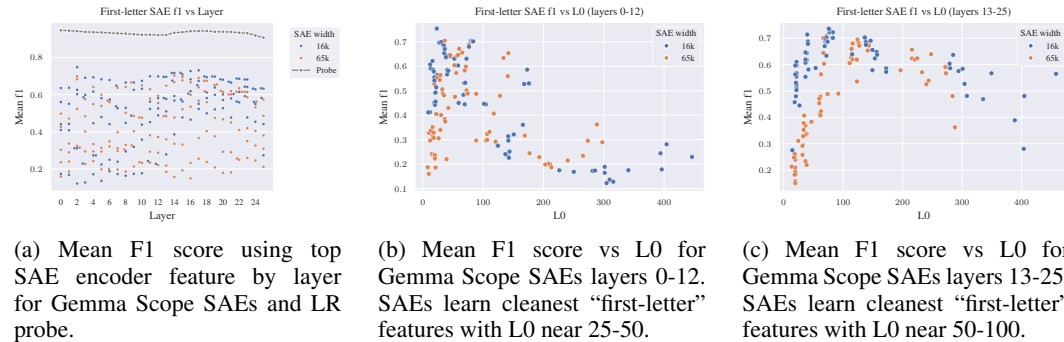

(a) Mean F1 score using top SAE encoder feature by layer for Gemma Scope SAEs and LR probe.

(b) Mean F1 score vs L0 for Gemma Scope SAEs layers 0-12. SAEs learn cleanest "first-letter" features with L0 near 25-50.

(c) Mean F1 score vs L0 for Gemma Scope SAEs layers 13-25. SAEs learn cleanest "first-letter" features with L0 near 50-100.

Figure 2: Comparison of F1 scores for first-letter classification tasks

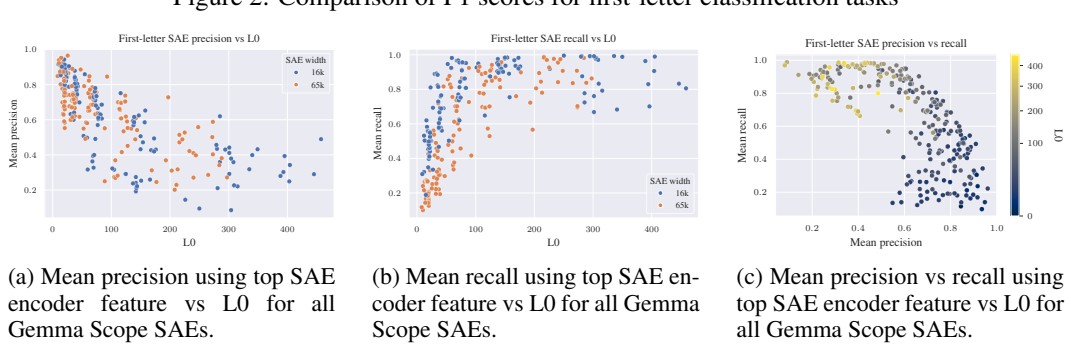

(a) Mean precision using top SAE encoder feature vs L0 for all Gemma Scope SAEs.

(b) Mean recall using top SAE encoder feature vs L0 for all Gemma Scope SAEs.

(c) Mean precision vs recall using top SAE encoder feature vs L0 for all Gemma Scope SAEs.

Figure 3: Precision and recall vs L0 for first-letter classification tasks

We also evaluate absorption on our own SAEs trained on Qwen2 0.5B (Yang et al., 2024) and Llama 3.2 1B (Dubey et al., 2024).

## 4  RESULTS

Our results are divided into three sections. First, we compare the performance of linear probes with SAE latents on recovering first-character information from model activations, showing that despite appearing to track first letter features, a wide variety of precision / recall is achieved. Second, we motivate our definition of feature absorption with a case-study, emphasizing how an absorbing feature can unexpectedly causally mediate first letter information whilst the first-letter latent (unexpectedly) fails to fire. Finally, we attempt to quantify feature splitting and feature absorption, showing that tuning of hyper-parameters may partially assist but not fully alleviate feature absorption.

### 4.1  DO SAEs LEARN LATENTS THAT TRACK FIRST LETTER INFORMATION?

We compare the performance of LR probes with the performance of the SAE latent whose encoder direction has highest cosine similarity with the probe, resulting in 26 "first-letter" latents. We observed that for each probe, there was clearly one or at most a couple of outlier SAE latents with high probe cosine similarity. Full plots of cosine similarity vs letter are shown in Appendix A.5.

We also experimented with using k=1 sparse probing to identify SAE latents (Gurnee et al., 2023), and find this gives similar results. Further comparison of k=1 sparse probing and encoder cosine similarity is explored in Appendix A.4.

We observe wide variance in the performance of Gemma Scope SAEs at the first-letter identification task, but no SAE matches LR probe performance. We show the mean F1 score by layer as well as the F1 score of the LR probe in Figure 2a. We further investigate the F1 score of these SAE encoder latents as a function of L0 and SAE width in Figures 2b and 2c.

Whether or not an SAE learns a clear "first-letter" latent for each letter is highly dependent on L0, with low L0 SAEs tending to learn high-precision low-recall latents, and high L0 SAEs learning

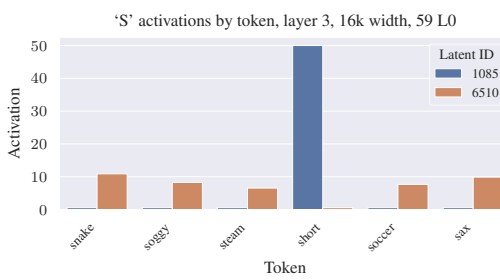 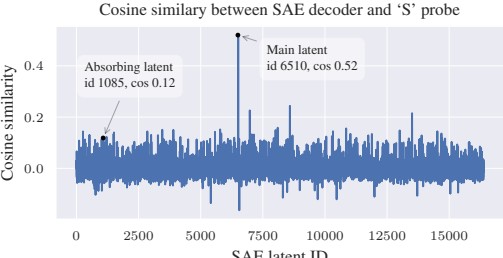

(a) Layer 3, L0=59 SAE feature activations for tokens that start with "S". The core "starts with S" feature, 6510, fails to activate on the token _short. The "short"-token aligned feature 1085 activates instead.

(b) Cosine similarity between layer 3, L0=59 SAE decoder and the "starts with S" probe. The main "Starts with S" latent, 6510, is clearly visible and highly probe-aligned.

Figure 4: SAE activations and cosine similarity for "starts with S" features.

low-precision high-recall latents (Figure 3). We caution drawing conclusions about an "optimal" L0 from these plots, as we find further variance when broken-down by letter, shown in Appendix A.5.

## 4.2 WHY DO SAE LATENTS UNDERPERFORM?

The Gemma Scope layer 3, 16k width, 59 L0 SAE has a latent, 6510, which appears to act as a classifier for "starts with S", achieving an F1 of 0.81. However, this latent fails to activate on some tokens the probe can classify, and which the model can spell, such as the token _short.

Figure 4a shows a sample prompt containing a series of tokens that start with "S", and the activations of top SAE latents by ablation score for these tokens. The main "starts with S" latent, 6510, activates on all these tokens except _short. This SAE also has a token-aligned latent, 1085, which activates on variants of the word "short" (" short", "SHORT", etc...). The Neuronpedia dashboard (Lin & Bloom, 2023) for feature 1085 is shown in Appendix A.11. For the token _short, the main "starts with S" latent does not activate but the "short" latent activates instead.

Latent 1085 has a cosine similarity with the "starts with S" probe of 0.12, indicating it contains a component of the "starts with S" direction, although much smaller than the main "starts with S" latent. Cosine similarity of the SAE decoder with the "starts with S" LR probe is shown in Figure 4b. Interestingly, despite latent 1085 having only about $1/5$ the cosine similarity with the probe as the main latent 6510, we see it activates with about $5$ times the magnitude of latent 6510 on the _short token, thus contributing a similar amount of the "starts with S" probe direction to the residual stream.

We conduct an ablation experiment on the _short token, shown in Figure 5a, and see that latent 1085 has a dramatically larger ablation effect compared with all other SAE features. This suggests latent 1085 is causally responsible for the model knowing that _short starts with S.

Is it possible that the probe projection is not the causally important component of feature 1085? We conduct another ablation experiment, except now we remove the probe direction from feature 1085 via projection before ablation. The results of this ablation experiment are shown in Figure 5b. After removing the probe component from feature 1085, it no longer has a significant ablation effect. Thus we know the probe projection of feature 1085 is responsible for model behavior.

These experiments show the "starts with S" feature has been "absorbed" by the token-aligned latent 1085, likely along with other semantic concepts related to the word "short". After observing that the main "starts with S" latent 6510 activates on most tokens that begin with "S", it may be tempting to conclude this latent tracks the interpretable feature of beginning with the letter "S". However, this latent quietly fails to activate on the _short token, leading us to a false sense of understanding.

We call this phenomenon **feature absorption**. In feature absorption a seemingly interpretable SAE latent fails to activate on arbitrary positive examples, and instead the feature is "absorbed" into approximately token-aligned latents.

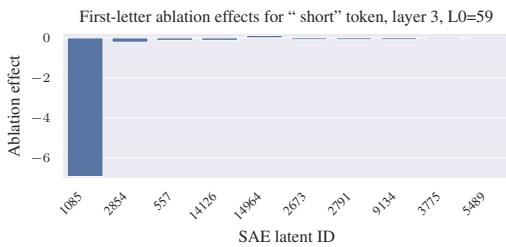 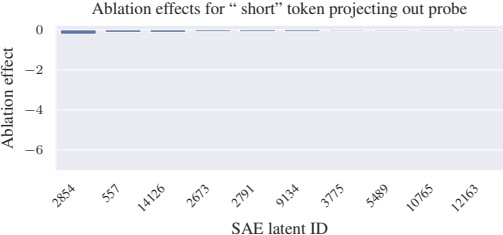

(a) Ablation effect for _short token, indicating that feature 1085, is responsible for the "starts with S" concept for the _short token. The main "starts with S" latent, 6510, does not activate on the _short token.

(b) Ablation effect for _short token after removing the probe direction from latent 1085 via projection. Latent 1085 no longer appears in the plot, indicating the strong ablation effect in Figure 5a is due to its component along the probe direction.

Figure 5: Ablation effects on _short token before and after projecting out the probe direction

| Latent 7112 | Latent 7657 |
|---|---|
| žda se napla==ću==je naknada | ==LC==, an aluminum boat |
| . E. Sø==li==, 20 | as ==LIFT== and ==LF==-Net. Once |
| a></==li==></ul | latter's sister ==Louise==, who in |

Table 1: Sample max activating examples for latents 7112 and 7657 for Gemma Scope 16k, layer 0, 105 L0 from Neuronpedia. The token where the SAE feature activates is highlighted in yellow. Latent 7112 appears to be a lowercase "L" starting-letter latent, and latent 7657 appears to be a corresponding uppercase "L" latent.

Feature absorption is likely a logical consequence of SAE sparsity loss. If a dense and sparse feature co-occur, absorbing the dense feature into a latent tracking the sparse feature will increase sparsity.

### 4.3 MEASURING FEATURE SPLITTING AND FEATURE ABSORPTION

**Feature splitting** A key phenomenon identified from previous studies of SAEs is feature-splitting (Bricken et al., 2023), where a feature represented in a single latent in a smaller SAE can split into two or more latents in a larger SAE. During our experiments, we found strong evidence of feature-splitting in the Gemma Scope SAEs.

For instance, in the layer 0, 16k width, 105 L0 SAE, we find two encoder latents (id:7112 and id:7657 [2]) which align with the "L" starting letter probe. Inspecting max activating examples, we see latent 7112 activates on tokens starting with lowercase "l", while 7657 activates on tokens starting with uppercase "L". Some activating examples for these latents are shown in Table 1.

Feature splitting like this is not necessarily problematic for interpretability efforts since the split features are still easily identifiable, and depending on the context it may be more useful to have either a single "starts with L" latent or a pair of "starts with uppercase / lowercase L" latents.

We measure feature splitting using k-sparse probing (Gurnee et al., 2023) on SAE activations. If increasing the k-sparse probe from $k$ to $k + 1$ causes a significant increase in probe F1 score, then the additional SAE latent provides a meaningful signal, and the combination of these $k + 1$ latents is likely a feature split. In the example of the uppercase "L" and lowercase "l" split, a k-sparse probe with $k = 2$ trained on both these these features should predict "starts with letter L" much better than either feature on its own. Figure 6a shows F1 vs K for letters "L" and "N". The "L" k-sparse probe shows a significant jump in F1 score moving from k=1 to k=2 corresponding to feature splitting, while the F1 score for the "N" k-sparse probe is relatively constant.

We detect feature splitting by measuring whether increasing $k$ by one causes a jump in F1 score by more than threshold $\tau$. We set $\tau = 0.03$ after manually inspecting latents with various thresholds. Figure 6b shows feature splitting vs L0 for all 16k and 65k width Gemma Scope SAEs.

---

[2]https://www.neuronpedia.org/list/cm0h1n2mt00019jdk274owq9e

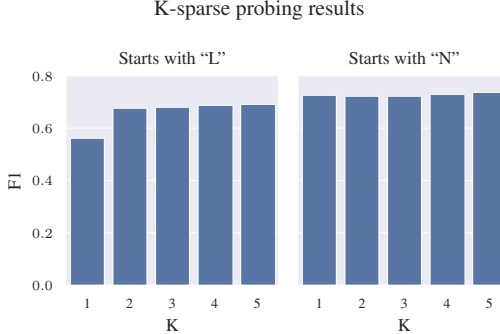
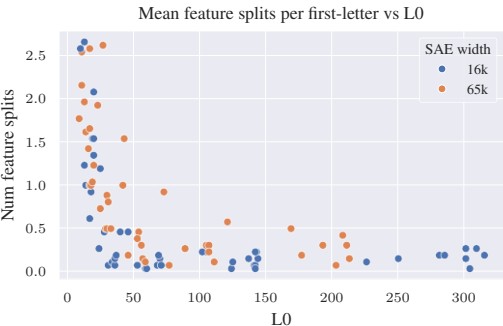

(a) K-sparse probing results for letters "L" and "N", layer 0, 16k width, 105 L0. "L" shows a significant improvement in F1 between k=1 and k=2 corresponding to feature splitting.

(b) Mean number of feature splits per letter on the first-letter spelling task by L0. Feature splitting occurs more frequently with higher sparsity.

Figure 6: Feature splitting

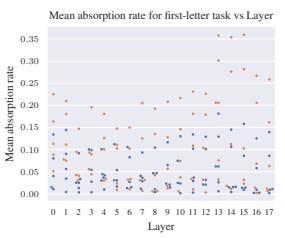
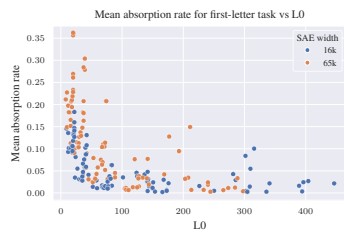
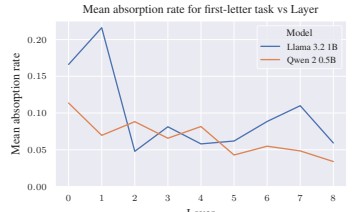

(a) Mean feature absorption rate vs layer on the first-letter task, Gemma Scope 16k and 65k SAEs. We do not see an obvious pattern in absorption rates by layer.

(b) Mean feature absorption rate vs L0 on first-letter task, Gemma Scope 16k and 65k SAEs. Wider and more sparse SAEs demonstrate higher rates of absorption.

(c) Mean feature absorption rate vs layer on the first-letter task on Llama 3.2 1B and Qwen 2 0.5B models, standard L1 loss SAE architecture, layers 0-8.

Figure 7: Feature absorption rates

**Feature absorption** The single latent or a set of traditional feature split latents that seem to act as a classifier for a human-interpretable feature like "starts with S" fail to fire in a seemingly arbitrary number of cases. What fires instead are approximately token-aligned latents with small but positive alignment with the LR probe. We say these latents are absorbing the feature.

We quantify the extent to which feature absorption occurs with the metric **feature absorption rate**. We first find $k$ feature splits for a first-letter feature using a k-sparse probe. We then find false-negative tokens that all $k$ feature-split SAE latents fail to activate on, but which the LR probe correctly classifies, and run an integrated-gradients ablation experiment on those tokens. The ablation effect finds the most causally important SAE latents for the spelling of that token. If the SAE latent receiving the largest negative magnitude ablation effect has a cosine similarity with the LR probe above 0.025, and is at least 1.0 larger than the latent with the second highest ablation effect, we say that feature absorption has occurred. These thresholds were chosen from manual inspection of the data. We then calculate feature absorption rate as below:

$$\texttt{absorption\_rate} = \frac{\texttt{num\_absorptions}}{\texttt{lr\_probe\_true\_positives}}$$

If there are more than 200 false negative per letter, we randomly sample 200 samples to estimate the number of absorptions. We see that absorption rate increases with higher sparsity and higher SAE width. Lower L0 likely pushes the SAE to absorb dense features like spelling information across multiple latents, thereby increasing feature sparsity. Feature absorption rate vs L0 for Gemma Scope SAEs layers 0-17 is shown in Figure 7b. Absorption rate by letter is shown in Appendix A.10. We also train our own set of standard L1 loss SAEs on the first 8 layers of Qwen2 0.5B (Yang et al., 2024) and Llama 3.2 1B Dubey et al. (2024). In Figure 7c we show that absorption occurs in these SAEs as well.

Our metric cannot capture absorption past layer 17 in Gemma 2 2B since we rely on ablation experiments to be certain the absorbed feature causally mediates model behavior. Past layer 17, attention has already moved the starting letter information from the source token into the final token position, so any ablations on the source token past layer 17 have little effect. This is a limitation of our absorption metric - we rely on ablation to be certain of the causal impact of absorbed features on model behavior, but this limits the layer depth our metric can be applied. We discuss this further in Appendix A.9.

Our absorption metric is not perfect, and is likely an under-estimate of the true level of feature absorption. We only consider absorption to have occurred if a single SAE latent has a much larger ablation effect than all other latents, and if the main SAE latents for a feature do not activate at all. Our metric will not capture multiple absorbing latents activating together, or the main latents activating but very weakly. Regardless, we feel our metric is a reasonable conservative baseline.

# 5 RELATED WORK

## 5.1 APPLICATIONS OF PROBES AND SAEs FOR MODEL INTERPRETABILITY

Probing methods have often been used to extract interpretable information from language models. However, the existence of such a representation does not guarantee that the model relies on this representation in its computation graph (Elazar et al., 2021).

Prior work has shown that many human-interpretable concepts in LLM activations are represented as linear directions in activation space, known as the linear representation hypothesis (Elhage et al., 2022; Park et al., 2024). Li et al. (2023) used non-linear probes to recover board representations from a transformer trained on Othello scripts ("OthelloGPT"). However, Nanda (2023) later showed that linear representations were not only recoverable but also editable.

Recent work has focused on applying SAEs to extract human-interpretable explanations of model internals. Karvonen et al. (2024) investigated how SAEs represent board states of Chess and Othello, and introduce a coverage metric using ground-truth features to evaluate the quality of SAE latents. This is very similar to our technique for evaluating SAE latents on a known task.

Other work has noted poor recall / precision of SAE latents compared to known proxies (Olah et al., 2024b; Kissane et al., 2024; Templeton et al., 2024). We build on this work by evaluating a large number of Gemma Scope SAEs to demonstrate how precision / recall is mediated by sparsity, and also offer a possible explanation of low recall due to feature absorption.

## 5.2 CHARACTER-LEVEL INFORMATION IN LANGUAGE MODELS

The ability of LLMs to learn character-level information from ostensibly character-blind tokens has been studied by various scholars, though no clear mechanism has yet been established. Kaushal & Mahowald (2022) trained MLPs from the embedding layers of GPT-J as probes for each letter in the alphabet, again finding good performance that implied character-level information was represented, but did not look into the model internals to explain how these representations were being used. In a follow-up work, Watkins & Bloom (2023) demonstrated that even linear probes on the embedding layers perform comparably well to MLPs in extracting character-level information.

In contrast to the above approaches, we train various probes for each character on multiple layers, and compare with SAE latents for a variety of layers.

## 5.3 DECOMPOSING SAE LATENTS

The phenomenon of SAEs of different sides splitting a feature into various smaller latents was first described in Bricken et al. (2023), which noted that different SAE widths and sparsities induce latents of different granularity, with wider SAEs often learning more specific variants of features.

Bussmann et al. (2024) find that by training an SAE on the decoder of another SAE, a technique called Meta-SAEs, it is possible to break down a single SAE latent like "Einstein" into subcomponents like "German" and "Physicist" and "starts with E". Meta-SAEs may provide a promising future research direction to overcome the feature absorption phenomenon we describe in this paper.

## 6 DISCUSSION

**Interpretability of SAE latents**   In this work, we use a simple first-letter identification task to investigate whether SAEs extract monosemantic and intrepretable features, and how this is affected by varying hyperparameters like SAE size, sparsity, or layer. We find that the SAE latents we investigated were not interpretable and that varying the sparsity or the size of the SAE did not meaningfully change this.

One may argue it is unreasonable to judge an SAE latent against a linear probe directly optimized to perform the same classification task and that it has been established that sometimes unsupervised methods can surprise us (Nanda, 2023). However, we argue that for a latent to be considered interpretable, its behavior should match what one would reasonably expect the latent to be doing after inspecting its activation patterns. In our experiments, we use SAE latents that do appear to perform first letter classification, and then evaluate how well they perform this task. We validate as well that these latents causally mediated model performance on the first-letter task in Appendix A.3. We are convinced that these latents should reasonably be considered "first-letter" latents and that their performance on the first-letter identification task is a valid measure of their interpretability.

**Feature absorption**   In trying to understand why SAE latents fail to match the performance of LR probes, we identified a form of feature splitting we call "feature absorption". Feature absorption may be particularly problematic for SAEs because it creates an interpretability illusion where we believe we have found an interpretable latent, but absorption induces lower recall by creating clear false negatives to the mainline interpretation of the latent. For example, we may believe we have found a SAE latent which tracks deceptive behavior in the model, but due to feature absorption, there may be many cases where that latent fails to fire. This lower recall poses problems for methods which rely on using SAEs to find sparse circuits (Marks et al., 2024), as the number of latents needed to characterize model behavior may be much larger than expected. We find that feature absorption happens even in high-recall latents, so this is not only a problem for low L0 SAEs and appears to be a more fundamental issue.

We hypothesize that feature absorption is a consequence of co-occurrence between sparse and dense features. If a dense feature like "starts with letter D" always co-occurs with a more sparse feature like "dogs", the SAE can increase sparsity by absorbing the "starts with D" feature into a "dogs" latent. We explore this further in Appendix A.7, where we show that feature absorption occurs when training an SAE in a toy setting with features that co-occur together.

It remains to be seen if we can predict or identify excepted instances where a feature "should have activated" but does not activate due to absorption. One promising direction is meta-SAEs, a novel method for decomposing SAE latents and may decompose absorbed features (Bussmann et al., 2024). One interpretation of our results is that competition may exist between "latents" and "meta-latents" for activation on particular examples and that re-allocation of examples between SAE latents enables SAEs to interpolate between different possible decompositions.

**Future Work**   A primary goal of future work should be to secure further external validity of our findings. This could include finding examples of feature absorption in SAEs trained on other models, with other architectures, or finding examples of feature absorption unrelated to character identification. We expect it should be possible to demonstrate feature absorption in a toy model by mixing dense features with sparse features that always co-occur with these dense features.

We hope as well this investigation may lead to research into solutions, particularly those involving Meta-SAEs (Bussmann et al., 2024), to solve or mitigate feature absorption. Anoter possible solution may be attribution dictionary learning (Olah et al., 2024a).

**Limitations**   Our feature absorption metric requires having ground-truth knowledge of true labels to first train a LR probe, whereas many features of interest in a LLM lack such clear-cut ground-truth labels. Our metric uses ablation effect to ensure absorbed features causally mediate model behavior, but therefore cannot be easily used in final model layers.

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

# A  APPENDIX

## A.1  GLOSSARY OF TERMS

**Sparse Autoencoders (SAEs):**  Neural networks trained to reconstruct their input while enforcing sparsity in their hidden layer. In the context of this paper, SAEs are used to decompose the dense activations of language models into more interpretable features.

**SAE error term:**  When inserting a SAE into the computation path of the model, errors in SAE reconstruction will propagate to later parts of the model and can change the model output. We refer to the error as the SAE error term, and corresponds to the difference between the SAE output and the original SAE input activation. Marks et al. (2024) introduced the idea of adding this error term back to the SAE output to ensure that the SAE does not change model output.

**Latent:**  We refer to neurons in the hidden layer of a SAE as latents to avoid overloading the term "feature". This is in contrast to earlier work which used the term "feature" to refer to both human-interpretable concepts and SAE hidden layer neurons.

**Feature:**  We use the term "feature" to refer to an idealized human-interpretable concept that the model represent in its activations and which a SAE latent may or may not represent.

**Monosemantic:**  Referring to a feature or representation that corresponds to a single, clear semantic concept. In the context of SAEs, a monosemantic feature would ideally capture one interpretable aspect of the input.

**Interpretable:**  A latent being interpretable is not well defined in the field, making it difficult to ensure that different authors mean the same thing when referring to SAE interpretability. When we refer to a SAE latent as being interpretable in this work, we mean that it should behave in line with how it appears to behave after inspecting its activation patterns. If an SAE latent appears

to track a feature X by a reasonable inspection of its activations but has subtle deviations from this behavior in reality, we say this is not interpretable. We thus measure interpretability via classification performance when a latent appears to be a classifier over some feature.

**Feature dashboard:**    A dashboard showing activation patterns and max-activating examples for a SAE latent. Feature dashboards are commonly used to interpret the behavior of an SAE latent.

**Neuronpedia:**    A platform, `https://neuronpedia.org`, which hosts feature dashboards for popular SAEs (Lin & Bloom, 2023).

**Token-aligned latent:**    A latent which seems to roughly fire on variants of the same token. For instance, a "Snake" token-aligned latent may fire on the tokens "Snake", "SNAKE", "_snakes", etc...

**Feature splitting:**    A phenomenon in SAEs introduced by Bricken et al. (2023), where a SAE latent tracks a general feature in a narrow SAE, but splits into multiple more specific SAE latents in a wider SAE. For instance, a latent tracking "starts with L" in a narrow SAE may split into a latent tracking "starts with capital L" and a latent tracking "starts with lowercase L" in a wider SAE.

**Feature absorption:**    A problematic form of feature splitting where a SAE latent appears to track an interpretable feature, but that latent has seemingly arbitrary exception cases where it fails to fire. Instead, an approximately token-aligned feature "absorbs" the feature direction and fires in place of the main latent.

**Circuit:**    In the context of neural network interpretability, a circuit refers to a subgraph of neurons or features within a neural network that work together to perform a specific function or computation. The study of circuits aims to understand how different components of a neural network interact to process information and produce outputs.

**Linear probe:**    A simple linear classifier (typically logistic regression) trained on the hidden activations of a neural network to predict some property or task. Used to assess what information is linearly decodable from the network's representations.

**K-sparse probing:**    A variant of linear probing where only the k most important features (as determined by some selection method) are used to train the probe. This helps identify which specific neurons or features are most relevant for a given task.

**Ablation study:**    An experimental method where a component of a system (in this case, a neuron or feature in a neural network) is removed or altered to observe its effect on the system's performance. This helps determine the causal importance of the component.

**Integrated gradients (IG):**    An attribution method that assigns importance scores to input features by accumulating gradients along a path from a baseline input to the actual input. In this paper, it's used as an approximation technique for ablation studies.

**In-context learning (ICL):**    A paradigm where a language model is given examples of a task within its input prompt, allowing it to adapt to new tasks without fine-tuning. Often used with few-shot learning techniques.

**Residual stream:**    In the context of transformer architectures, the residual stream refers to the main information flow that bypasses the self-attention and feed-forward layers through residual connections.

**Logits:**    The raw, unnormalized outputs of a neural network's final layer, before any activation function (like softmax) is applied. In language models, logits typically represent the model's scores for each token in the vocabulary.

**Activation patching:** An interpretability technique where activations at specific locations in a neural network are replaced or modified to observe the effect on the network's output. This helps in understanding the causal role of different parts of the network in producing its final output.

## A.2 HOW GOOD IS GEMMA-2 ON CHARACTER IDENTIFICATION TASKS?

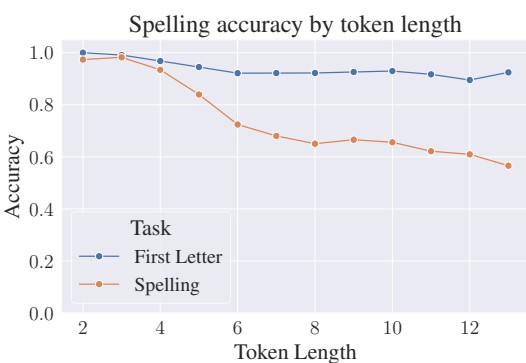

Figure 8: Baseline performance for Gemma-2-2B on first-letter identification and full-token spelling by token length.

We evaluate how well can Gemma-2-2B identify the first letter or all the letters in a token (spelling the full token). We evaluate the accuracy of the model on all tokens in the LR probe validation set with a prompt containing 10 in-context examples selected at random from the full vocabulary. Our results are shown in Figure 8.

We see that performance on the first-letter identification task is high throughout token length, while the full-word spelling performance decreases as the length of the token increases.

## A.3 INTERVENING ON THE FIRST LETTER

If the model is using the identified SAE latents for predicting the first letter we should also be able to change what first letter it predicts just by changing the activations. For this experiment we use the SAE latents most cosine similar with the LR probe for the true first letter and for a new randomly selected letter. We take the intermediate activations of Gemma-2-2B in the residual stream and encode them using the SAE. Then we zero out the activation of the SAE latent associated with the original letter and change the activation of the SAE latent associated with the new letter into the average activation it has on tokens starting with this new letter.

Editing works better with latents from the narrower 16k SAE compared to the 65k, with the best L0s in the 75-150 range. This corresponds to the observed pattern of these SAE latents having higher F1 scores for classification. We report the results in Figure 9. The best SAEs on the layers 7-9 can achieve a substantial replacement, but note that the averages hide variance across individual tokens, where some get edited completely and others get unaffected. The edit success also varies based on the true first letter and the random new letter; for illustration we show a breakdown by letter for two specific SAEs in layer 7 in Figure 10.

## A.4 PROBE COSINE SIMILARITY VS K=1 SPARSE PROBING

The first step when searching for a SAE feature that acts as a first-letter classifier involves searching for SAE feature which best acts as a classifier. In Figure 2, we achieve this by first training a LR probe on the first-letter task and using cosine similarity between that probe and the SAE encoder to find the best feature for the first-letter task. We also investigated using k-sparse probing with k=1 to select the best SAE feature instead. This involves training a linear probe with L1 loss and selecting the feature with the highest positive weight from the probe.

We find that both k=1 sparse probing yield nearly identical results, as seen in Figures 11 and 12. Additionally Figure 13 shows the cosine similarity of the LR probe with each SAE feature by letter

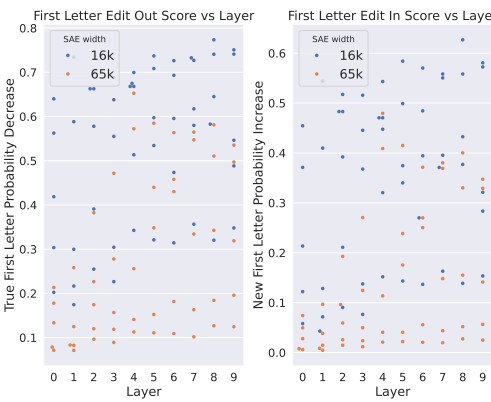 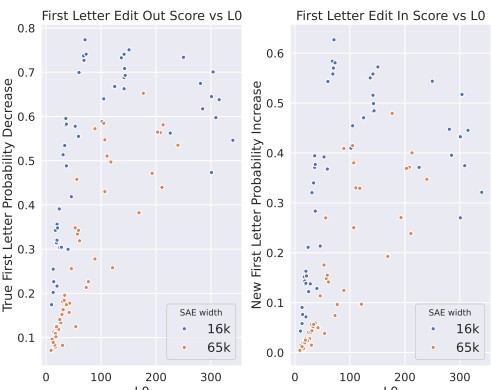

(a) Comparing success in editing out the true first letter and making the model predict a randomly selected new letter across layers 0-9 for all 16k and 65k Gemma Scope SAEs.

(b) Comparing the edit success with the top SAE feature across all L0s for 16k and 65k widths across layers 0-9. The best performance seems to be occurring for L0 between 75 and 150.

Figure 9: Comparison of Edit success by Layer and L0

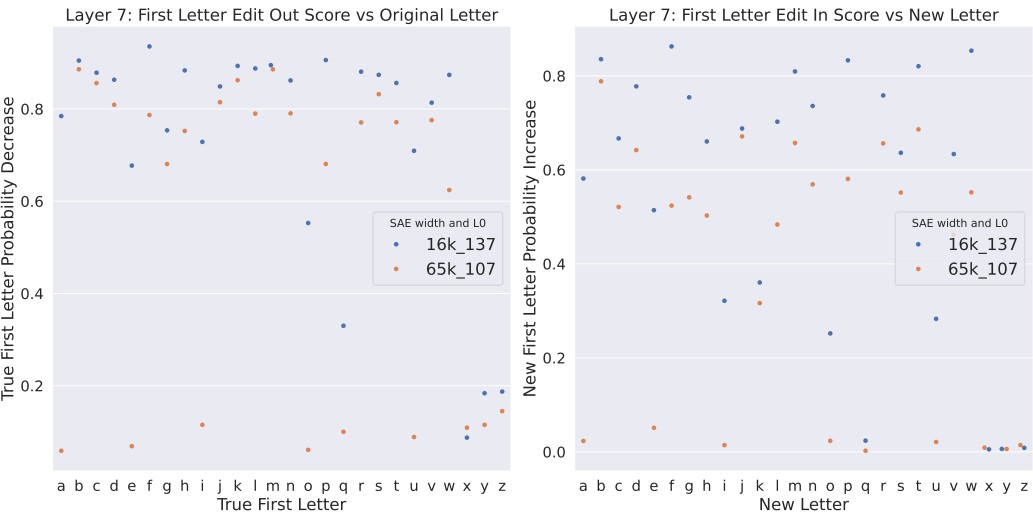

Figure 10: Comparing the edit success broken down by the letter at layer 7 for two SAEs; SAE width 16,000 and L0 of 137 and SAE width 65,000 and L0 of 107. For each original letter we draw a sample of 100 tokens and average the decrease in probability of the correct first letter and increase in probability of a new random letter.

for the canonical Gemma Scope layer 0 16k width SAE. In most cases there is an obvious probe-aligned feature. Likely any reasonable method of feature selection will find the same feature for these cases. We thus decided to use cosine similarity between the SAE encoder and a LR probe as our selection criteria for single SAE features as this is a simpler metric and less computationally intensive to compute.

### A.5 PRECISION, RECALL, AND F1 SCORE FOR THE FIRST-LETTER TASK

We evaluated precision, recall, and F1 score for the first-letter classification task, and found that the precision and recall vary depending on the L0 of the SAE. Low L0 SAEs learn high precision, low recall features, while high L0 SAEs learn low precision, high recall features. These results are shown in Figure 14. We thus chose to use F1 score as our core metric in this paper to balance precision and recall as many if the SAEs we tested have extreme values in either precision or recall.

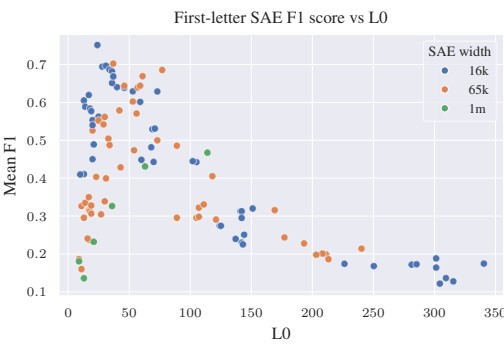 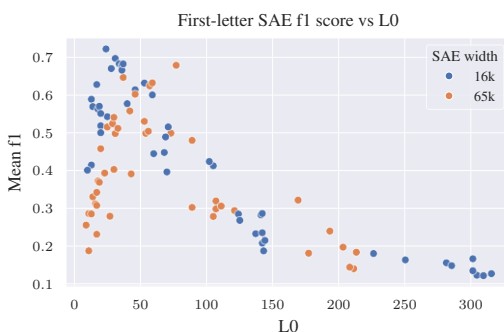

(a) Mean F1 score on first-letter classification task using top SAE encoder feature by cosine similarity with the LR probe vs L0 for all Gemma Scope SAEs layers 0-9.

(b) Mean F1 score on first-letter classification task using k=1 sparse probing to select the SAE feature for the first-letter classification task vs L0 for all Gemma Scope SAEs layers 0-9.

Figure 11: Comparison of LR probe cosine similarity and k=1 sparse probing vs l0

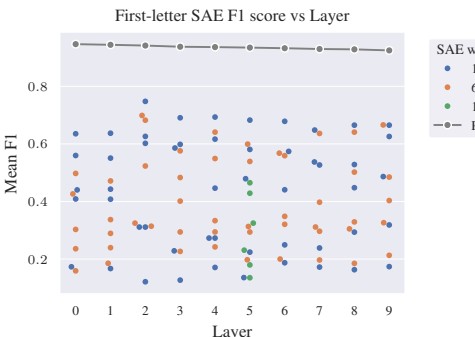 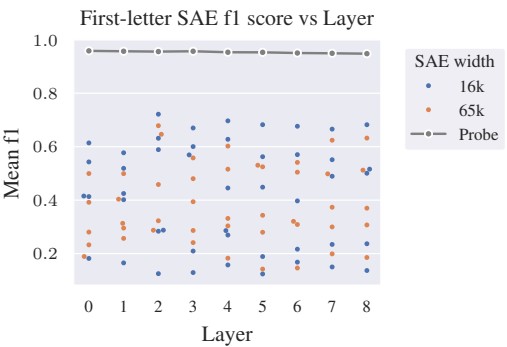

(a) Mean F1 score on first-letter classification task using top SAE encoder feature by cosine similarity with the LR probe vs layer for all Gemma Scope SAEs layers 0-9.

(b) Mean F1 score on first-letter classification task using k=1 sparse probing to select the SAE feature for the first-letter classification task vs layer for all Gemma Scope SAEs layers 0-9.

Figure 12: Comparison of LR probe cosine similarity and k=1 sparse probing vs layer

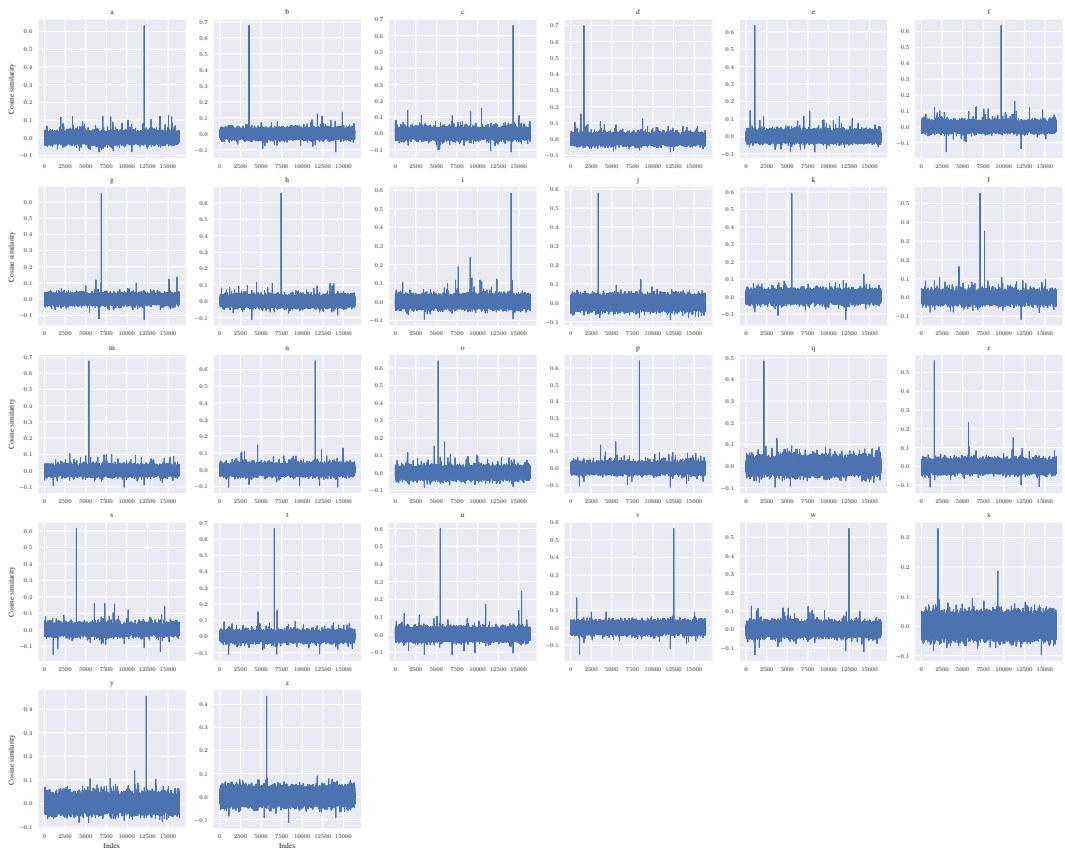

Figure 13: Decoder cosine similarities with the LR probe by letter, Gemma Scope 16k layer 0 l0=105. Most letters have one or two obvious SAE features which align with the probe.

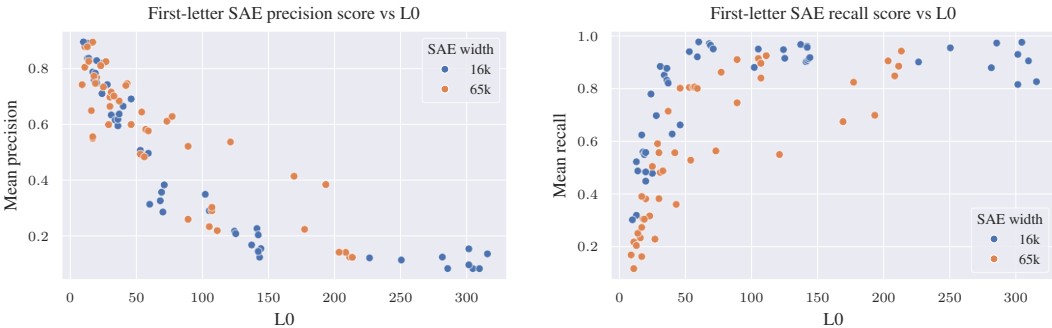

(a) Mean precision on first-letter classification task vs L0 for all Gemma Scope SAEs layers 0-9. Features are selected via k=1 sparse probing

(b) Mean recall on first-letter classification task vs L0 for all Gemma Scope SAEs layers 0-9. Features are selected via k=1 sparse probing

Figure 14: Comparison of precision and recall vs l0

While it may appear that there is an optimal L0 from looking at aggregate statistics across letter, we find that breaking down the F1 vs L0 plot by letter reveals that the optimal L0 appears different for different letters, with low frequency letters like z actually having the best F1 score at the lowest L0, while other letters instead have an optimal L0 around 30-50. Figure 15 shows these results broken down by letter.

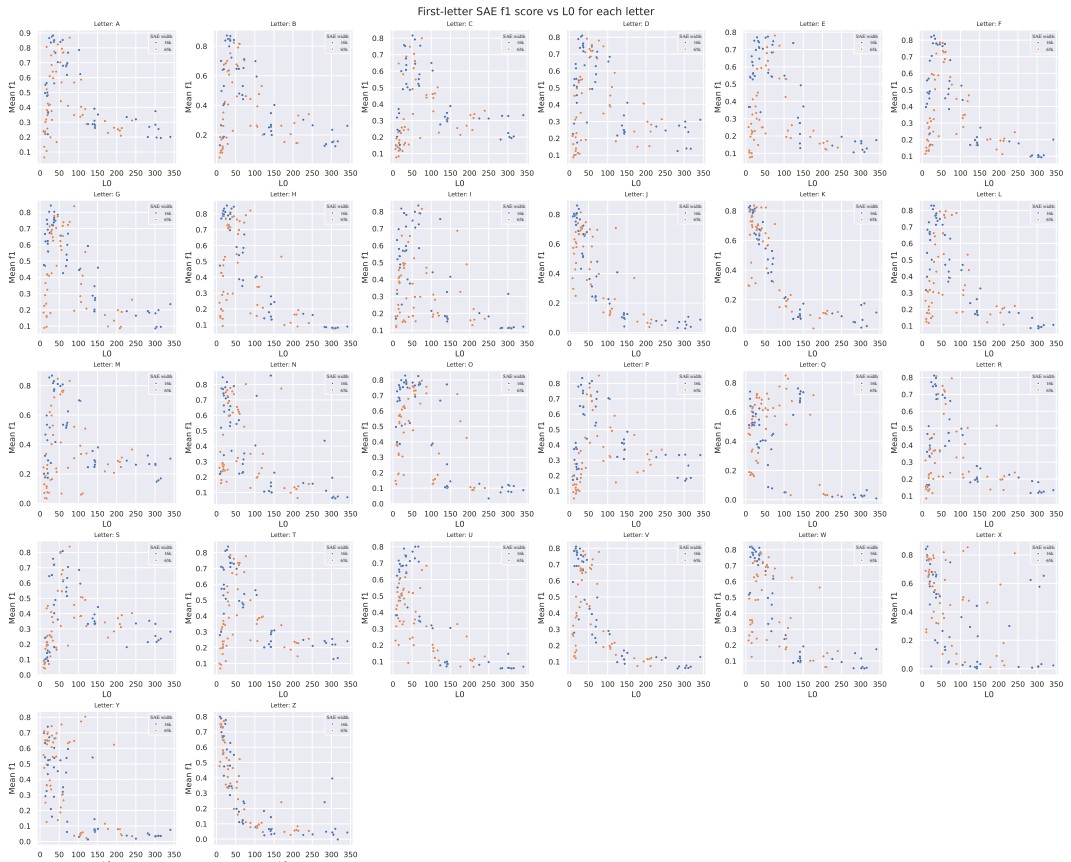

Figure 15: F1 vs L0 by letter. SAE features are picked using k=1 sparse probing.

## A.6 SAE TRAINING

We train SAEs on the first 8 layers of Qwen2 0.5B (Yang et al., 2024) and Llama 3.2 1B (Dubey et al., 2024) using the SAELens library (Joseph Bloom & Chanin, 2024). The SAEs are all trained with identical hyperparameters of L1 coefficient of 2.5 and 500M tokens. The Qwen2 0.5B SAEs all have L0 between 25 and 50 and explained variance between 0.77 and 0.83. The Llama 3.2 1B SAEs have L0 between 27 and 110, and explained variance between 0.74 and 0.89.

## A.7 TOY MODELS OF FEATURE ABSORPTION

We hypothesize that absorption is due to feature co-occurrence combined with the SAE maximizing sparsity. When two features co-occur, for instance "starts with S" and "short", the SAE can increase sparsity by merging the "starts with S" feature direction into a latent tracking "short" and then simply not fire the main "start with S" latent. This means firing one feature instead of two, and thus increasing sparsity.

We show that feature co-occurrence does indeed cause absorption by constructing a toy setting with four true features and an SAE with four latents.

**Setup** Our initial setup consists of 4 true features, each randomly initialized into orthogonal directions with a 50 dimensional representation vector and unit norm. We control the base firing rates of each of the 4 true features. Unless otherwise specified, the feature fires with magnitude 1.0 and stdev 0.0. We train a SAE with 4 latents to match the 4 true features using SAELens (Joseph Bloom & Chanin, 2024). The SAE uses L1 loss with l1 coefficient 3e-5, and learning rate 3e-4. We train on 100,000,000 activations. Our 4 true features have the firing rates shown in Table 2.

|  | Feature 0 | Feature 1 | Feature 2 | Feature 3 |
|---|---|---|---|---|
| Firing rate | 0.25 | 0.05 | 0.05 | 0.05 |

Table 2: Base firing rate for features in toy experiment.

We use this setup for the following reasons:

- This is a very easy task for a SAE, and it should be able to reconstruct these features nearly perfectly.
- Using fully orthogonal features lets us see exactly what the L1 loss term incentivizes without worrying about interference from superposition.

**Independently firing features** When the true features fire independently, we find that the SAE is able to perfectly recover these features as shown in Figure 16.

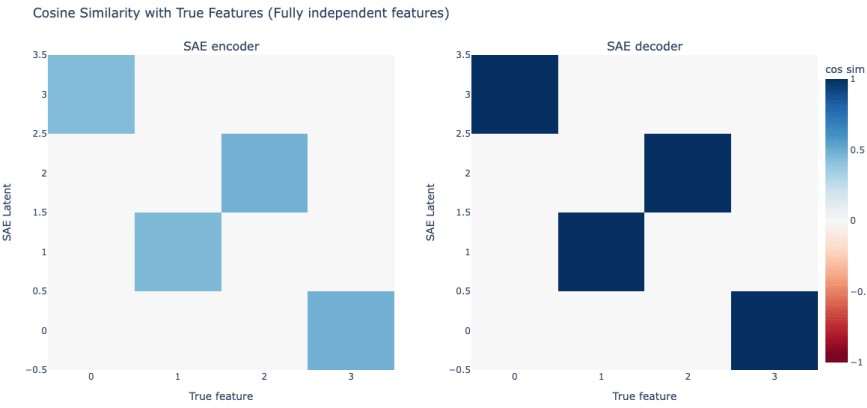

Figure 16: When features fire independently, the SAE learns exactly one latent per feature, and the decoder perfectly reconstructs the feature.

We see the cosine similarity between the true features and the learned encoder, and likewise with the true features and the decoder. The SAE learns one latent per true feature. The decoder representations perfectly match the true feature representations, and the encoder learns to perfectly segment out each feature from the other features.

**Co-occurrence causes absorption** Next, we modify the firing pattern of feature 1 so it fires only if feature 0 also fires. However, we keep the overall firing rate of feature 1 the same as before, firing in 5% of activations. Features 2 and 3 remain independent.

Figure 17 shows the encoder and decoder cosine similarities with the true features in the co-occurrence setup. Here, we see a clear example of feature absorption. Latent 0 has learned a perfect representation of feature 0, but the encoder has a hole in its recall. Latent 0 fires if feature 0 is active but not feature 1. This is exactly the sort of gerrymandered feature firing pattern we saw in real SAEs for the starting letter task - the encoder has learned to stop the latent firing on specific cases where it looks like it should be firing. In addition, we see that latent 3, which tracks feature 1, has absorbed the feature 0 direction. This results in latent 3 representing a combination of feature 0 and feature 1. We see that the independently firing features 2 and 3 are untouched - the SAE still learns perfect representations of these features.

A.8    METRIC CHOICE FOR ABLATION STUDIES

To determine the causal effect of SAE latents on the first-letter identification task, we use a metric, $m$, which measures the logit of the correct letter minus the mean logit of all incorrect letters. Our

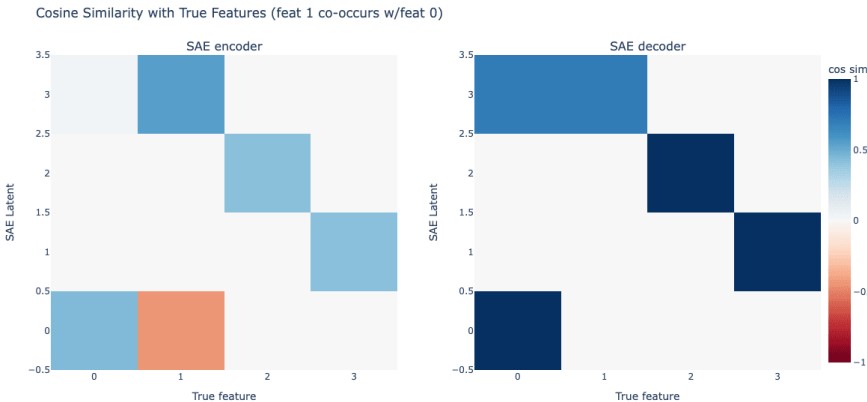

Figure 17: When features 0 and 1 co-occur, we see absorption in the SAE encoder and decoder latents which track features 0 and 1.

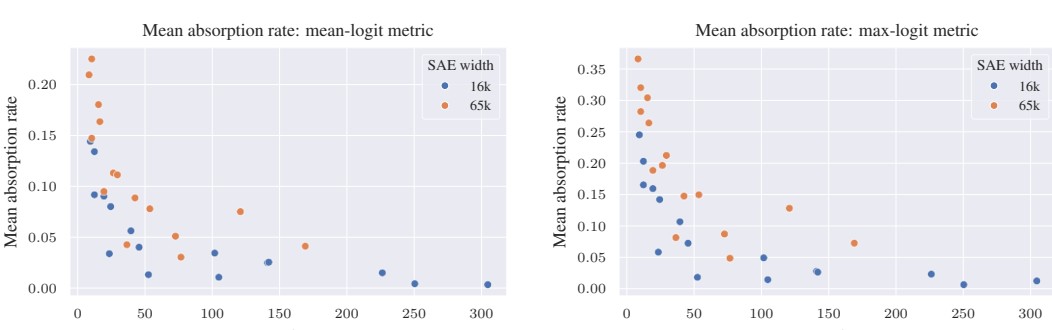

(a) Absorption rate using the mean version of the absorption metric, Gemma Scope layers 0-3.

(b) Absorption rate using the max version of the absorption metric, Gemma Scope layers 0-3.

Figure 18: Comparison of absorption rates using the max and mean versions of the absorption metric.

metric is defined below, where $g$ refers to the final token logits, $L$ is the set of uppercase letters, and $y$ is the uppercase letter that is the correct starting letter:

$$m = g[y] - \frac{1}{|L| - 1} \sum_{l \in \{L \setminus y\}} g[l]$$

This metric is chosen to detect changes in the confidence of the model in predicting the correct letter relative to the mean reference class of other letters. This should capture changes in the model's confidence in predicting the correct logit.

This is not the only metric that could be chosen, and an argument can be made that we should subtract the max of all incorrect letter logits rather than the mean of all incorrect letter logits. The max form of this version of the metric is shown below:

$$m_{max} = g[y] - \max_{l \in \{L \setminus y\}} g[l]$$

This second form using a max can also account for the case where the logits of the model shift from being confident in the correct answer to instead being confident in an incorrect answer while leaving the logits of the correct answer the same.

In practice, we expect that ablating an absorbing latent should cause the model to become less confident in the correct answer, so the difference between these two forms of the metric should yield similar results.

We calculate the mean absorption rate for Gemma Scope SAEs layers 0-3 in Figure 18 using both versions of this metric. The overall shape of the curve is nearly identical between these two choices of metrics. The mean version of the metric, which is used in this paper, results in a slightly more conservative estimate of absorption rate.

We consider our absorption score to be a rough estimate of the true absorption rate and thus consider either the mean or the max version of the logit diff metric to be valid for evaluating absorption.

### A.9 CAUSAL INTERVENTIONS AND ABSORPTION

In this work, we rely on causal interventions like ablation experiments to verify that SAE latents have a causal impact on model behavior. In these experiments for spelling tasks, we set up an ICL prompt to elicit spelling information from the model, for instance the ICL prompt below:

```
tartan has the first letter: T
mirth has the first letter: M
dog has the first letter:
```

In this ICL prompt, we would apply an SAE and train LR probes on the _dog token position, and expect that the model will output the token _D. When we intervene on the _dog, we can track the causal changes to model outputs by applying a metric to the output logits, e.g. checking how our intervention increases or decreases the _D logit relative to other letters.

We use these interventions as part of our absorption metric to ensure that when we claim that "absorption" is occurring, we verify that the absorbing feature has a causal impact on model outputs. This is stronger evidence than only noting a cosine similarity between the absorbing feature, but this means that our absorption metric cannot classify absorption at later model layers.

During a LLM forward pass, the model first collects relevant information on a token in that token position, and attention heads then move relevant information from earlier tokens to later tokens (Geva et al., 2023; Meng et al., 2022). If we assess ablation effect at layers after which model attention has already pulled relevant information from the subject tokens into the final output token, the ablation effect will be 0. For Gemma 2 2B on the first-letter spelling task, we find this movement of first-letter spelling information occurs around layer 18.

Figure 19 shows an activation patching experiment (Meng et al., 2022) on a sample first-letter spelling prompt. In this experiment, we see that near layer 18 the model moves first-letter spelling information from the subject token to the prediction token.

As a result, our feature absorption metric will not function past layer 18 in Gemma 2 2B, and we thus focus on layers 0-17 for our analysis of feature absorption. We believe that feature absorption is still occurring in SAEs past layer 18, but we lose the ability to make causal claims that the absorbing features are used by the model to make predictions. Given that this paper is trying to highlight the existence of feature absorption, we felt it is more important to have a metric which is robust and has the backing of causal analysis but which cannot be used at all model layers. Future work may make a different trade-off and choose a feature absorption metric which can work at all model layers, for instance relying only on cosine similarity between absorbing features and a LR probe to determine absorption.

### A.10 ADDITIONAL PLOTS

In this section, we include additional plots that are too large to fit in the main body of the paper.

### A.11 FEATURE DASHBOARDS

We include feature dashboard screenshots from Neuronpedia for some prominent latents mentioned in this work. Figure 21 shows a dashboard for Gemmascope layer 3, latent 1085, which is a token-aligned latent firing on variations of the word _short and we find absorbs the "starts with S"

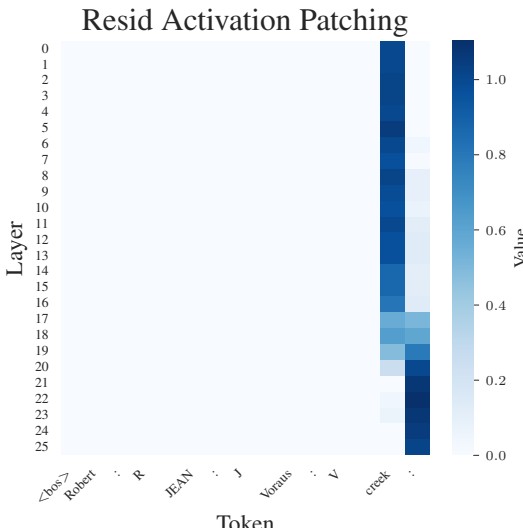

Figure 19: Residual stream attribution patching for a sample first-letter spelling prompt, Gemma 2 2B. After around layer 18, model attention moves the relevant spelling information from the source token to the prediction location.

direction. Figure 22 shows latent 6510 from the same layer which should be the main "starts with S" latent.

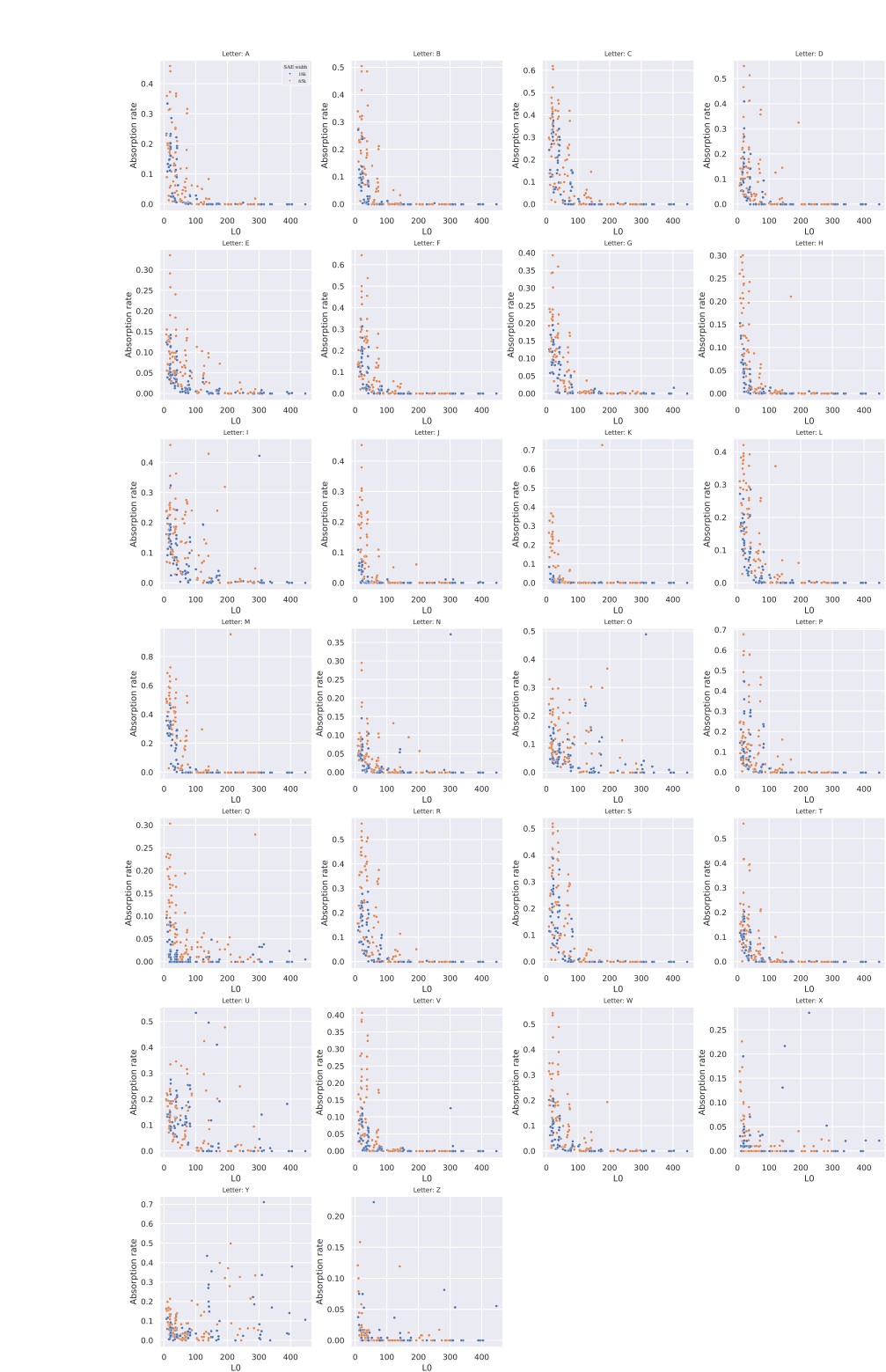

Figure 20: Absorption rate vs L0 by letter, layers 0-17. We see a wide variance in which letters are absorbed by which SAEs.

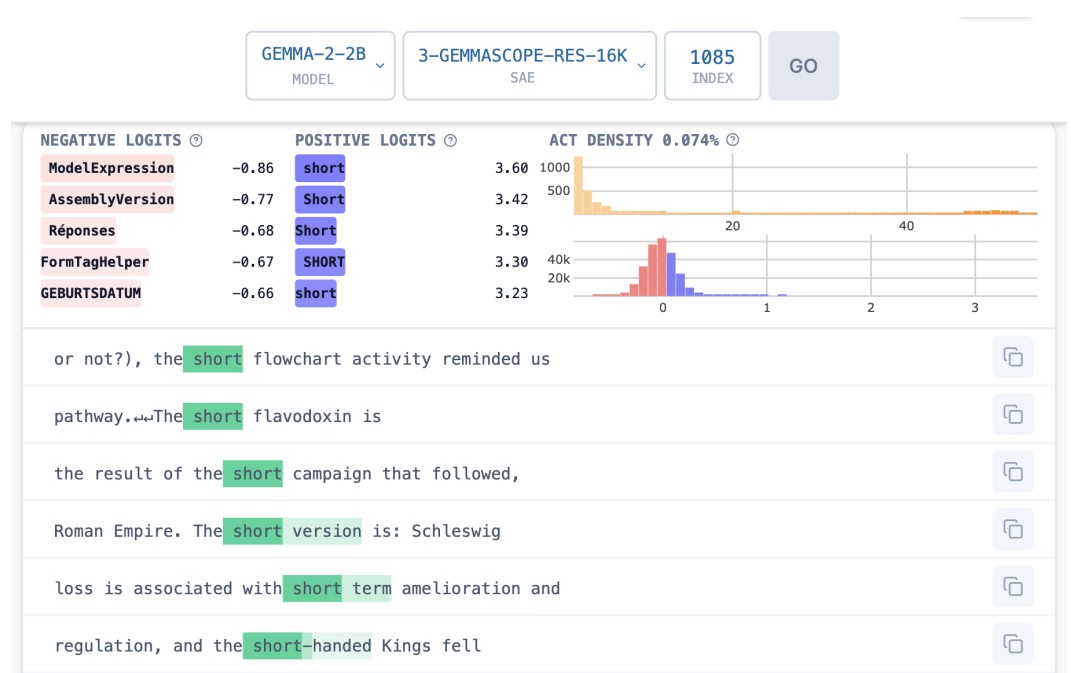

Figure 21: Neuronpedia dashboard for Gemma Scope layer 3, latent 1085. This latent is a token-aligned latent for _short tokens. This latent absorbs the "starts with S" direction.

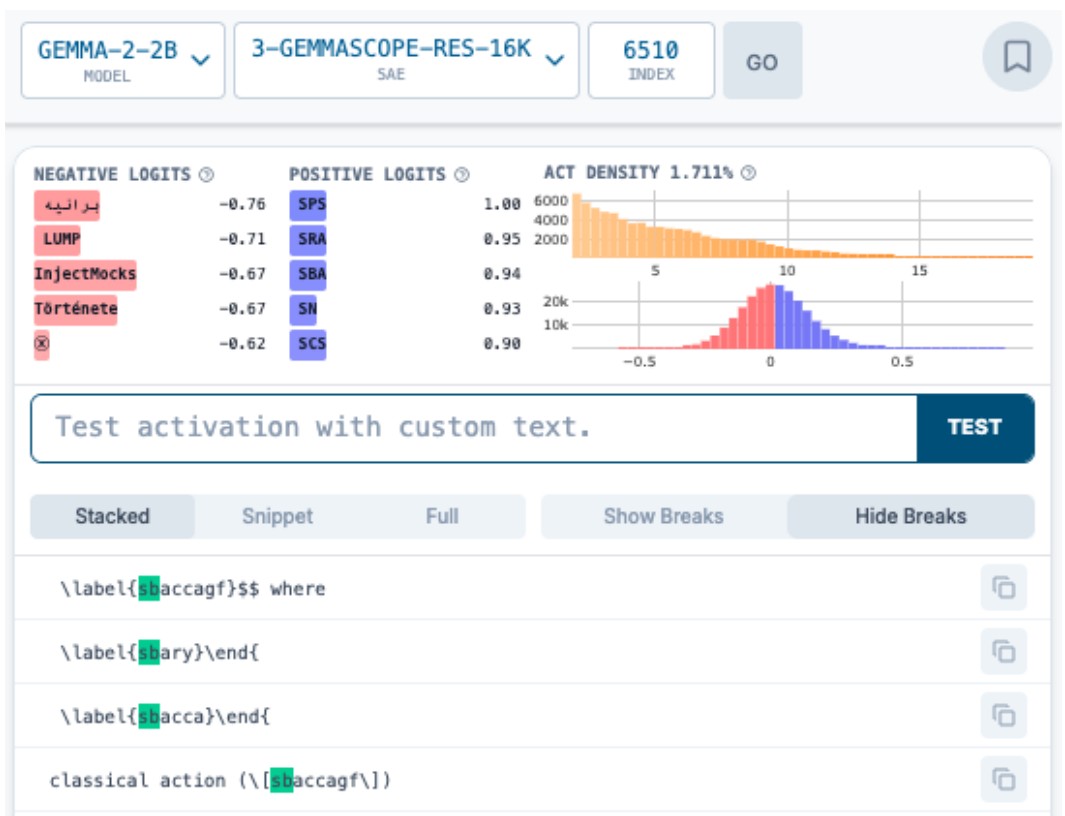

Figure 22: Neuronpedia dashboard for Gemma Scope layer 3, latent 6510. This latent should be the main "starts with S" latent.

