# OpenReview forum: "A is for Absorption: Studying Feature Splitting and Absorption in Sparse Autoencoders"
_ICLR.cc/2025/Conference — Submitted to ICLR 2025_

### Official Review · Reviewer_x2j6 · 2024-10-30

**Soundness:** 3
**Presentation:** 3
**Contribution:** 3
**Rating:** 8
**Confidence:** 4

**Summary:**

This paper documents and measures a particular form of feature splitting in the Gemma Scope family of sparse autoencoders (SAEs), which the authors call feature absorption. The prototypical case of feature absorption studied is when a latent is a high-precision classifier for a property like "starts with the letter L", but has poor recall due to other latents "absorbing" individual tokens, so that the main latent's activations are given by the rule "starts with the letter L, except for the tokens laser, lions, [several other exceptions]".

**Strengths:**

The paper explores a natural task, first letter classification, and does so well. It establishes that this information is linearly available in the model (by training an LR probe with high F1-score), and that SAEs find latents that align with the probe direction (Figure 4b and especially Figure 13), and that these latents are often good classifiers, though not as good as probes (Figure 2a).

The paper provides a novel perspective to a documented issue of feature splitting in SAEs. It provides good theoretical justification of why feature absorption is likely to happen, and why it provides challenges to interpretability.

**Weaknesses:**

The paper could do more to demonstrate that feature absorption happens as described, namely that:
1) There is a latent which is a high-precision classifier for "starts with letter [x]".
2) There is a particular subset of tokens which start with this letter but which the latent fails to activate on.
3) There are separate token-aligned latents that individually classify those tokens.
4) If that subset of tokens is removed, the latent is a high-precision, high-recall classifier for "starts with the letter [x], except for that subset of tokens".

The authors demonstrate 1) and 2) well in sections 4.1 and 4.3 respectively, but never show 3) or 4) besides a single example in section 4.2. The authors could address 3) by adding an additional step to the experiment in section 4.3, in which they confirm that "the SAE latent receiving the largest negative magnitude ablation effect" is a token-classifying latent.

Lines 148-157: The metric described can fail at its stated goal to "[measure] the propensity of the model to choose the correct starting letter as opposed to other letters" because averaging logits can interact poorly with the softmax activation function. For example, if "A" is the correct letter, but the model logits are g[A]=10, g[B]=25, g[C]=g[D]=...=g[Z]=0, then the model would produce B with near-certainty. But the provided metric yields m=10-(25+0+0+...+0)/25=10-1=9, the same value as if g[A]=9, g[B]=...=g[Z]=0. This metric would be more convincing if the average were replaced by a max, or if it were a softmax over letter tokens.

The authors only study a single family of SAEs on a single language model. Alternative SAE architectures may not demonstrate feature absorption.

**Questions:**

**Questions**

1. The concept of "approximately token-aligned" latents is central to claim 3 (lines 69-71), but is not defined anywhere. How is token-alignment defined, and is it quantified or measured in these experiments?

2. When the F1 score of an SAE is computed, as in Figure 2 (lines 169-172), how are SAE activations converted into a binary classification? Is the classification "does/doesn't activate". Is a non-zero threshold used? If so, how is that threshold chosen?

3. In Figure 7b, when the mean absorption rate is \~35%, is that indicating that for roughly 35% of tokens, there is a latent absorbing that token? The vocabulary size of Gemma-2B is \~256k (https://arxiv.org/html/2408.00118v1#:~:text=Vocab%20size-,256128,-256128) and 35% of that is \~90k. If that is the case, how are 90k tokens absorbed into 65k latents?

4. In figure 13, why is the high cosine similarity in the .4-.6 range, and not closer to 1? This range of cosine similarities is suggestive of learning the "true" feature mixed with a second, orthogonal feature.

5. It seems that the paper usually uses cosine similarity of the LR direction with the encoder direction, but Figures 4 and 13 use decoders instead of encoders. Why is that?

6. What does Algorithm 1 (Lines 116-127) return? One ablation effect per token per latent? Or are these averaged in some way?

**Other Comments**

If it is permitted for authors to make revisions before the final submission, there are several small changes that could improve the quality of the paper:

- Lines 94-107: the paper should reference (Lieberum et al., 2024) since that is the exact architecture of SAE being studied.

- Line 100: Although ReLU is often used in SAE architectures, it would be better to write a generic activation function in this equation, especially considering that the SAEs studied in the paper use JumpReLU, not ReLU.

- Line 118 and Line 147: The description "include the SAE error term" is ambiguous in the text. It is clarified in the glossary, and by the glossary's citation to (Marks et al., 2024), but the main paper neither cites (Marks et al., 2024) here, nor links to the glossary. Is there a way to include hyperlinks to the glossary in the text itself?

- Line 205: There is a citation to (Gao et al., 2024) for sparse probing. It is likely supposed to cite (Gurnee et al., 2023) as in the background section. (Gao et al., 2024) use k-sparse SAEs, which are distinct from k-sparse probes and the SAE architecture studied in the paper.

- Lines 291-292: "The token where the SAE feature activates is highlighted in green" should be "yellow" since the highlight is yellow.

- Line 404: The paper says "it is difficult to apply [(Karvonen et al, 2024)] to existing SAEs trained on real LLM activations." One of the main contributions of (Karvonen et al, 2024) is very applicable to this paper, namely coverage (Section 3.2 in https://arxiv.org/pdf/2408.00113). Coverage for a given set of properties (in this case, "first letter identification") is defined by, for each property, finding the SAE latent and threshold which results in the best F1 score for classifying that property, then averaging those F1 scores across the properties. This is very similar to what is shown in Figure 2, though with a different selection rule for SAE latents. The paper could make reference to the similarity of this method to coverage.

- Line 429, possible typo: "..find that by training an SAE on the decoder *or* another SEE, a technique...", the word "or" should be "of".

- (Uncertain) The citation of "(Huben et al., 2024)" should instead refer to the paper as "(Cunningham et al., 2024)". The order of the authors is different between the OpenReview page (https://openreview.net/forum?id=F76bwRSLeK) and the paper itself (https://openreview.net/pdf?id=F76bwRSLeK). In such a case it is most likely proper to follow the order of authors in the paper.

**References**

Leo Gao, Tom Dupre ́ la Tour, Henk Tillman, Gabriel Goh, Rajan Troll, Alec Radford, Ilya Sutskever, Jan Leike, and Jeffrey Wu. Scaling and evaluating sparse autoencoders. arXiv preprint arXiv:2406.04093, 2024.

Adam Karvonen, Benjamin Wright, Can Rager, Rico Angell, Jannik Brinkmann, Logan Riggs Smith, Claudio Mayrink Verdun, David Bau, and Samuel Marks. Measuring progress in dic- tionary learning for language model interpretability with board game models. In ICML 2024 Workshop on Mechanistic Interpretability, 2024.

Wes Gurnee, Neel Nanda, Matthew Pauly, Katherine Harvey, Dmitrii Troitskii, and Dimitris Bertsi- mas. Finding neurons in a haystack: Case studies with sparse probing. Transactions on Machine Learning Research, 2023. ISSN 2835-8856. URL https://openreview.net/forum? id=JYs1R9IMJr.

Robert Huben, Hoagy Cunningham, Logan Riggs Smith, Aidan Ewart, and Lee Sharkey. Sparse autoencoders find highly interpretable features in language models. In The Twelfth International Conference on Learning Representations, 2024. URL https://openreview.net/forum? id=F76bwRSLeK.

Tom Lieberum, Senthooran Rajamanoharan, Arthur Conmy, Lewis Smith, Nicolas Sonnerat, Vikrant Varma, Ja ́nos Krama ́r, Anca Dragan, Rohin Shah, and Neel Nanda. Gemma Scope: Open Sparse Autoencoders Everywhere All At Once on Gemma 2, August 2024.

Samuel Marks, Can Rager, Eric J. Michaud, Yonatan Belinkov, David Bau, and Aaron Mueller. Sparse feature circuits: Discovering and editing interpretable causal graphs in language mod- els. Computing Research Repository, arXiv:2403.19647, 2024. URL https://arxiv.org/ abs/2403.19647.

---

> ### Author Response · Authors · 2024-11-17
>
> We thank the reviewer for their in-depth review and feedback of our paper. We have addressed all the points in the comments section in the current version of the paper on OpenReview.
>
> We address the weaknesses mentioned below:
> - While we usually see absorption occurring in token-aligned latents, it is not essential to our argument that the absorbing latent be token-aligned. For instance, if a latent represents “snakes and spiders” rather than just token variations on the word “spider”, absorption of “starts with S” can still occur. We even saw some cases where a latent that absorbs “starts with S” fires mostly, but not exclusively, on things starting with S. We have clarified this wording in the paper. We also created a streamlit app to help explore our data, mentioned in the abstract which can help to get a feel for absorbing latents.
> - We do not claim that without absorption, SAE latents would otherwise be perfect classifiers. Our claim is only that absorption is an issue that affects performance of SAE latents as classifiers, not that it is the only issue.
> - You bring up an interesting edge-case with the metric we use for measuring ablation effect. Our metric is chosen to capture the logit of the correct answer relative to all other plausible answers. In general, we are interested in the model moving from being certain that the next letter is A to being less certain or confused about what the next letter is. We do not expect that ablating a SAE latent would cause the model to increase an incorrect letter latent while keeping the correct letter constant, but we do agree that if that were to occur our metric would struggle to capture it. To test this, we evaluated both versions of the logit-diff metric on layers 0-3 of Gemma Scope and found the results to be very similar. We added these results and a discussion of the max version of the metric to Appendix A8. We consider either version of the metric to be valid when evaluating absorption as the choice of metric does not appear to significantly affect results.
> - We have trained a set of standard L1 loss SAEs on Qwen2 0.5B and Llama 3.2 1B models, and evaluated absorption on these additional SAEs and models. We included the results in the current version of the paper. We find that absorption occurs in all models and all SAE architectures, bolstering the case that absorption is due to the sparsity penalty rather than being an artifact of Gemma Scope SAEs or the Gemma 2 model. These results are in Figure 7c.
>
> We address the questions below:
> - Token-aligned latents just mean that a latent appears to match variations of a single token or word, so we would say a latent that fires on tokens like “snake”, “snakes”, “SNAKE” is token-aligned. This is just an observation of the sorts of latents we found absorption in the first-letter task to occur, rather than being fundamental to the argument of the paper. We clarified this in the paper, and added a definition of “token-aligned latent”.
> - For the F1-score of latents, we use the latent firing or not as the classifier. So a latent activating with any value larger than 0 is considered a positive classification decision.
> - If the absorption rate is 35%, this means that in 35% of the cases where the logistic regression (LR) probe correctly classifies the starting letter of a token as a true positive, the corresponding first-letter SAE latent(s) fail to activate. Instead, a different latent in the SAE fires instead containing the LR probe direction, and ablating that latent has a large negative effect on model output, indicating the latent is causally responsible for the model “knowing” the starting letter. There is no relation to the number of tokens in the vocabulary. It is possible for a single absorbing latent to fire on multiple tokens, for example (e.g. “snakes”, “_snake”, etc… could all be represented by the same latent, absorbing the “S” direction).
> - We do not know why the top probe-aligned SAE latents tend to have ~0.6 cos sim with the probe rather than a higher number. It could just be that in high-dimensional space, there are a large number of directions that will work as a classifier and the directions found by the probe and the SAE both operate successfully as a classification boundary with ~0.6 cos sim to each other. Investigating this further could be an interesting direction for future work.
>  - We find that using either the encoder or decoder cosine similarity for detecting probe-aligned latents both work equally well, as does using k=1 sparse probing. We find that plotting cosine similarity against the decoder tends to be slightly less noisy, but we do not feel there is a big difference between using the encoder or decoder cosine similarity for our analysis.
> - Algorithm 1 returns a single ablation effect per latent. This algorithm calculates the effect on the metric of ablating a single latent at a time, so each latent gets its own ablation score.

---

> > ### Comment · Reviewer_x2j6 · 2024-11-21
> >
> > Thank you for your response, and for the revisions to the paper. I especially appreciate the additional tests on other models (Qwen2 0.5 and Llama, and your toy model in Appendix A.7), and your tests with the alternative metric in Appendix A.8. In light of these improvements, I have updated by score to an 8/10.
> >
> > An additional direction I would love to see explored (possibly in a future work) is whether your tests on toy models answer my Question 4, about why cosine similarities are in the .4-.6 range. In particular, I am intrigued by your Figure 17, where due to absorption, the decoder direction is of the form [subset feature direction]+[superset feature direction], which has a cosine similarity of 0.5 with [superset feature direction]. Could this have occurred in Figure 13, where the letter-aligned latents are themselves absorbing latents for some hypothetical ur-features(s). For instance, perhaps there were ur-features for "starts with consonant" and "starts with vowel", and the letter-aligned latents are the absorbing features of those. Then their decoder directions would be of the form [starts with letter a]+[starts with vowel], which would have cosine similarity of 0.5 with the LR-probe direction [starts with letter a]. I would propose the following test for this:
> > 1. For each letter, take the decoder direction corresponding to it, and subtract off the component of the decoder corresponding to the LR probe direction (i.e., projecting onto the subspace perpendicular to the LR probe direction). In the example above, this would turn [starts with letter a]+[starts with vowel] into [starts with vowel], as long as those directions are orthogonal (which is plausible for this high-dimensional space).
> > 2. Find the cosine similarities between these 26 projected vectors.
> > 3. If there were ur-latents, these would show up as clusters of high-cosine-similarity vectors in this test.

---

> > > ### Author Response · Authors · 2024-11-23
> > >
> > > We thank the reviewer for their continued feedback and for increasing the score. The question about whether the reason for cosine similarities in the 0.4-0.6 range could be a result of a nested absorption pattern is indeed worth investigating. It could be possible that “starts with letter S” is itself in a hierarchical relationship with one or more other features in the model.
> > >
> > > Following your comment, we tried projecting out the probe direction from the top decoder latent for each letter on the Gemma Scope layer 3, 16k, 59 L0 SAE, and then compared the cosine similarities of these 26 latents to each other. We see many of these remaining vectors having cosine similarity around 0.25 with each other, although there are also many close to 0, and even some around -0.20. There are some interesting patterns in the cosine similarities, such as vowels having higher cosine similarity to each other, letters in the alphabet near each other having higher similarity, and phonetically similar letters like voiced/unvoiced variants (e.g. t/d, f/v, s/z) have higher similarity. This could indicate that there are multiple absorbing hierarchies at play (e.g. a “vowels” feature, phonetics feature, alphabet position feature, etc…), but would require a more thorough investigation across more SAEs to verify this for sure. This would be an exciting direction for future work, and we thank the reviewer for this suggestion.

---

> > > > ### Comment · Reviewer_x2j6 · 2024-11-23
> > > >
> > > > Thank you for running that experiment, and sharing those intriguing results! It would be exciting to see this further explored in a future work.

---

### Official Review · Reviewer_4XB2 · 2024-11-03

**Soundness:** 4
**Presentation:** 3
**Contribution:** 4
**Rating:** 8
**Confidence:** 4

**Summary:**

This paper proposes a new simplified setting for studying features discovered by Sparse Autoencoders (SAEs) trained on language model activations. It supposes that SAEs should discover features corresponding to the first letter of a given token, but finds that features which are predictive of a given letter occasionally fail and are 'absorbed' into the activation of another feature. They then measure and explore this phenomenon in the context of their setting.

**Strengths:**

- Describes an interesting setting in which to study feature disentanglement for natural language.
- Discovers a new phenomenon, 'feature absorbtion', which could pose a problem for desired future applications of Sparse Autoencoders.
- Setting allows for a thorough analysis of the failure cases for feature absorbtion and feature splitting, and said analysis was good and informative.
- Paper has a clear statement of the problem and good structure. Good use of the 'S'/'_short' example to help build intuition.

**Weaknesses:**

- It would be useful to explore the commonalities between different instances of absorbtion of SAE features, or otherwise find more ways to verify/expand upon the claim/statement that "feature absorption is likely a logical consequence of SAE sparsity loss".

**Questions:**

- Did you find any rules for predicting whether the feature activation for the first letter of a given word would be absorbed? (i.e. do you have any hypothesies/ways to differentiate situations in which feature absorbtion occurs or does not occur?)

---

> ### Author Response · Authors · 2024-11-17
>
> We thank the reviewer for their interest in the paper topic and their thorough review of the paper. We have thought a lot about why feature absorption occurs, and believe that absorption is a sparsity-maximizing solution when two features co-occur together. For instance, if a LLM has a feature for “starts with S” and a feature for “snakes”, then every time the “snakes” feature is active the “starts with S” feature will also be active. Since a SAE is incentivized to maximize sparsity, the SAE can abuse this fact to absorb the “starts with S” direction into the “snakes” latent, and thus only need to fire one latent on “snakes” rather than two.
>
> We expect absorption will occur any time there is a hierarchical relation like this, with a parent feature and child feature such that the parent feature is always active when the child feature is active. For instance, we would expect that things like “noun” would get absorbed into latents representing nouns, or “German” to get absorbed into any more specific latent about something German.
>
> We have been able to reproduce feature absorption empirically in a toy setting where we control the activation distribution of all latents, and can show that when two features co-occur absorption will occur. We expanded the discussion of this in the paper, and have also added a full exploration of the toy setting to Appendix A7 in the paper.

---

> > ### Comment · Reviewer_4XB2 · 2024-11-23
> >
> > Thank you for your response and extended discussion of co-occurance and its relation to absorbtion. The toy setting is very compelling. This has increased my confidence in the score I gave, and I'll update that accordingly.

---

### Official Review · Reviewer_1gVz · 2024-11-04

**Soundness:** 3
**Presentation:** 2
**Contribution:** 3
**Rating:** 6
**Confidence:** 3

**Summary:**

This paper investigates the interpretability of the extracted latent/features by Sparse Autoencoders. The authors found that the Sparse Autoencoder does extract monosemantic and interpretable features and changing the hyperparameters of the Sparse Autoencoder could not eliminate this.

**Strengths:**

As the large language model becomes increasingly popular, it is really important to get a good understanding of these techniques. This paper investigates a meaningful problem and attempts to provides some insights and points out some research directions, which should be very useful to the community.

#---------------------------------------#

After rebuttal, the authors add results with different SAE architectures and three different model types.

**Weaknesses:**

There are several weaknesses for this paper.
1) The observation and conclusion are merely based on one model and one task (first-letter identification task). Thus, it is really difficult to say how general and convincing these conclusions could be. I am afraid it is not safe to draw a solid conclusion based a single setting.
2)  The authors should elaborate their method better. The current version simply lists some background and then posts a pseudo code (algorithm) there. More descriptions should benefit readers to get a better understanding.
3) Some contents cause confusing instead of help. For example, the Figure1 is really confusing and hard to interpreter and even worse, the authors do not provide sufficient descriptions of it in the main paper.
4) The overall presentation is a little bit messy and it is really hard to follow the content.

**Questions:**

I list my concerns and questions in the "weaknesses" section.

---

> ### Author Response · Authors · 2024-11-17
>
> We thank the reviewer for taking the time to review our work and provide feedback.
>
> We would highlight that the core insight of the paper is introducing the idea of feature absorption and showing its appearance in real SAEs. Feature absorption is a degenerate sparsity-maximizing solution that SAEs can find when features (of the input) co-occur together, and induces SAE latents into failing to fire in some cases where the feature they seem to track has occurred.
>
> The bulk of the paper focuses on demonstrating that feature absorption occurs in real SAEs, even state-of-the-art SAEs like Gemma Scope. However, the argument about why absorption occurs is a logical result of the sparsity loss of SAEs. If “starts with S” and “snakes” are two features in a LLM, and every time “snakes” fires, “starts with S” also fires, a SAE can increase its sparsity by absorbing the “starts with S” direction into the “snakes” latent, thus requiring only one latent to fire rather than two. We have added a further exploration of this phenomenon using a toy setting with 4 features and an SAE with 4 latents in appendix A7, as the feature absorption phenomenon can be clearly seen even in this simple setting.
>
> We have also trained our own standard L1 loss SAEs on the Qwen2 0.5B and Llama 3.2 1B, and demonstrate that feature absorption occurs in these SAEs as well. We have added plots demonstrating this to the paper in Figure 7c. We hope this should alleviate any concerns related to only investigating the Gemma Scope SAEs on Gemma2 2B, as we now have results for two different SAE architectures across three model types.

---

> > ### Author Response · Authors · 2024-11-23
> >
> > Dear Reviewer,
> >
> > As the end of the discussion period is approaching, we want to ask if our response has addressed any of your concerns regarding the paper, or if you have any further questions or comments about our paper. We appreciate the time and effort put into the review process and would love the opportunity to discuss this further if our response didn't address your concerns. Thank you again for your time reviewing our paper.

---

> > ### Comment · Reviewer_1gVz · 2024-12-03
> >
> > Thanks so much for your time and efforts for the rebuttal. Thanks for providing more results with different SAE architectures and three different model types. Based on my understanding as well as the available reviews, I will modify my rating. But I still think a much better elaboration about the method as well as a better presentation (for example, adding more detailed explanation of Figure 1 in the main paper) could benefit readers.

---

### Official Review · Reviewer_S6kU · 2024-11-05

**Soundness:** 3
**Presentation:** 3
**Contribution:** 3
**Rating:** 8
**Confidence:** 3

**Summary:**

This paper explores the use of Sparse Autoencoders (SAEs) to decompose activations of Large Language Models (LLMs) into interpretable latent features. It addresses two main questions: the degree to which SAEs yield monosemantic (single-meaning) and interpretable latents, and how adjustments to the sparsity and size of SAEs impact interpretability. Through a controlled first-letter identification task with complete ground-truth labels, the authors find more nuanced insights compared to previous studies.

**Strengths:**

1. This paper studies an interesting problem, that is, learned sparse features can become less understandable by "absorbing" token-aligned features.
2. understanding how width and sparsity of SAEs affects its training performance can be helpful for future SAE training.
3. the paper is written in an easy-to-understand way.

**Weaknesses:**

1. It would be valuable to explore practical examples where insights from feature absorption can enhance interpretability. For instance, analyzing the first letters of all tokens associated with specific latent activations might offer new explanations for features that previously seemed opaque. This approach could reveal subtle patterns in latent activation, aiding in the interpretation of challenging features.
2. This work presents a specific form of feature absorption, but it raises the question of whether other variations might exist. Are there additional contexts or scenarios where feature absorption manifests differently, potentially impacting interpretability in distinct ways? Identifying these cases could deepen our understanding of the phenomenon and refine the strategies needed to address it.

**Questions:**

Same as above

---

> ### Author Response · Authors · 2024-11-17
>
> We thank the reviewer for their interest in the topic of our paper, and for the kind words about the paper being easy to understand. We agree that absorption may provide a lens through which to further examine previously uninterpretable latents that may in-fact be absorbing a number of co-occuring features, thus reducing their interpretability. We are excited about future work that can help us interpret these latents or even break absorbing latents back into interpretable components. We also suspect there may be more problematic feature distributions beyond simple co-occurrence that SAEs may find degenerate solutions to. We hope to examine these topics in future work.

---

> > ### Comment · Reviewer_S6kU · 2024-11-18
> >
> > Thanks for your reply. I have read through the reply and will keep my positive rating after considering the rebuttal.

---

### Public Comment · ~Neel_Nanda1 · 2024-11-16
**Paper Thoughts**

Speaking as an uninvolved mechanistic interpretability researcher, I consider this paper a valuable contribution to the literature, and think that it has improved my understanding of sparse autoencoders. Feature absorption is not something I'd thought about explicitly before, but in hindsight feels obvious, and I think has useful implications for strategic directions in future SAE research. To me the key takeaway is as follows:

The core motivation of sparse autoencoders is that we want to find interpretable concepts in an unsupervised way, don't know how to optimise for interpretability, and so instead optimise for the proxy of sparsity. And so a crucial question is how good a proxy sparsity is. Until recently, we didn't have much evidence of it being an issue, and I mostly ignored it. Feature absorption seems like the crispest example yet of how sparsity is an imperfect proxy:

Whenever you have two concepts A and B, where B is a strict superset of A (ie if A is present then B is definitely present) it will come apart. The example in this paper is pairs like A="this is the word elephant" and B="this token starts with E", but this is a very general point. The interpretable solution is to learn feature_1=A and feature_2=B, but when A is present, two features fire! So the sparsest solution is to absorb feature_2 into feature_1: feature_1=A and feature_2=(B excluding A).

Once the problem is demonstrated once, it becomes pretty clear that we should expect it to happen everywhere, whenever there are hierarchical pairs of concepts. I consider this a valuable contribution because it helps direct future SAE research: we need to either find a way to optimise less for sparsity, so feature absorption isn't incentivised; try a totally different approach; or show that the costs of absorption (and other failures of sparsity) are low enough to be worth paying.

(Disclosure: I was not at all involved in this paper, but I do know the authors and likely have some positive bias due to that. No one asked me to write this)

(Note also: I don't feel confident in the norms here, so feel free to ignore this comment if these kinds of thoughts are not welcome! I don't see other researchers doing this, but I figure that as ICLR has made a deliberate choice to allow public comments during the rebuttal process, they likely want this kind of public feedback from uninvolved researchers, so long as it's constructive)

---

### Author Response · Authors · 2024-11-17

We thank the reviewers for the time reviewing the paper and for their feedback on our work. We are grateful for the positive comments supporting the importance of the work to the field and the interest in potential follow-up work.

To address feedback from reviewers, we have made the following changes to the paper:
- We trained our own standard L1 loss SAEs on the Qwen2 0.5B and Llama 3.2 1B, and demonstrate that feature absorption occurs in these SAEs as well. We have added plots demonstrating this to the paper in Figure 7c. We hope this should alleviate any concerns related to only investigating the Gemma Scope SAEs on Gemma2 2B, as we now have results covering two different SAE architectures and three model types.
- We have been able to reproduce feature absorption empirically in a toy setting where we control the activation distribution of all latents, and can show that when two features co-occur absorption will occur. We expanded the discussion of this in the paper, and have also added a full exploration of the toy setting to Appendix A7 in the paper. This should help bolster the argument that feature absorption is a logical consequence of the sparsity penalty in SAEs when features co-occur together.
- We have made writing improvements to the locations in the paper pointed out in reviews, and feel this has helped improve the clarity and quality of the paper further as well.

---

### Public Comment · ~Neel_Nanda1 · 2025-02-18
**Why on earth was this rejected?**

I find the AC's decision to reject this paper deeply perplexing. There was an overwhelming and unanimous consensus in favour of accepting the paper from reviewers, at 3 accepts and a weak accept. Notably, of the 11,500 submissions to ICLR, this was the *third* highest rated rejected submission, and higher rated than 90% of accepted papers. I think that given reviews this strong, an extreme reason is required to justify overturning it and the reasons given for rejection seem wholly insufficient.

More importantly, rather than a case of shared poor reviewer judgement, this is a genuinely high quality paper, that has meaningfully added to the field, and helped shape how I think about my research. This decision is an unusually poor application of area chair discretion, and I hope this paper is resubmitted to a conference where it gets an AC with better judgement.

---

### Meta-Review · Area_Chair_xZjR · 2024-12-21

**Metareview:**

Sparse Autoencoders (SAEs) have become a popular tool for interpreting neuron activation behaviors in large language models (LLMs). This paper questions the effectiveness of sparsity regularization in SAEs by identifying the phenomenon of feature absorption—a behavior where overlapping features are absorbed into a single latent representation, degrading interpretability. The authors use a controlled first-letter identification task to analyze feature absorption, demonstrating its prevalence and persistence across varying SAE sizes and sparsity levels. They identify sparsity penalties in SAEs as the root cause of this phenomenon.

Multiple reviewers appreciated the identification and analysis of feature absorption, describing it as a meaningful contribution to the understanding of SAEs. The use of controlled settings and diverse models (e.g., Gemma Scope, Qwen2, Llama) provides robust empirical validation.

Concerns from the reviewers:

- Limited Real-World Applicability: The study focuses primarily on the first-letter identification task, limiting its generalizability to broader real-world tasks.

- Unexplored Variants of Absorption: Reviewer S6kU and others noted that additional manifestations of feature absorption or scenarios where absorption may behave differently were not explored, leaving gaps in understanding.

- Presentation Issues: Multiple reviewers criticized the paper’s overall clarity and organization, pointing out ambiguous descriptions and confusing visuals (e.g., Figure 1).

**Recommendation**

While there is considerable interest in the identification of feature absorption, as reflected in reviewer and public comments, and all reviewers recommended acceptance, the paper’s shortcomings are substantial:

1) Its focus on a toy problem raises question on the generalizability and practical relevance of its findings.

2) there is no rigorous theoretical characterization of the phenomenon, limiting the depth of its contributions.

3) it has limited attempts at addressing the identified problem. For example, if the issue is with hierarchical features then methods from hierarchical sparse regularization (see e.g. [a]) would be a natural alternative to (unstructured) sparse coding?

4) the lack of clarity in the presentation raises the concern that the paper may not be broadly accessible to the broad audience of ICLR. In particular, reviewer 1gVz raised multiple concerns on clarity of the presentation but there was little effort during the discussion phase to address those.

Given all these, I’d recommend a weak rejection while encouraging the authors to improve their manuscript along the lines above to substantiate their interesting findings.

[a] Proximal Methods for Hierarchical Sparse Coding, JMLR 2011

**Additional Comments On Reviewer Discussion:**

Concern: Exploration of alternative absorption behaviors; Exploration of practical examples where insights from feature absorption can enhance interpretability.

Response: The authors acknowledged this gap and suggested it as a direction for future work.

Concern: The observation and conclusion are merely based on one model and one task

Response: The authors conducted additional experiments on Qwen2 and Llama models; however it still is limited to the same task.

Concern: Lack of clarity and presentation issues from Reviewer 1gVz.

Response: No explicit response and reviewer still strongly calls for better presentation at the end.

Other concerns are mostly satisfactorily addressed during the discussion phase.

---

### Decision · Program_Chairs · 2025-01-22

Reject